# ESTIMATING INFORMATION FLOW IN DNNS

## ABSTRACT

We study the evolution of internal representations during deep neural network (DNN) training, aiming to demystify the compression aspect of the information bottleneck theory. The theory suggests that DNN training comprises a rapid fitting phase followed by a slower compression phase, in which the mutual information $I(X;T)$ between the input $X$ and internal representations $T$ decreases. Several papers observe compression of *estimated mutual information* on different DNN models, but the true $I(X;T)$ over these networks is provably either constant (discrete $X$) or infinite (continuous $X$). This work explains the discrepancy between theory and experiments, and clarifies what was actually measured by these past works. To this end, we introduce an auxiliary (noisy) DNN framework for which $I(X;T)$ is a meaningful quantity that depends on the network's parameters. This noisy framework is shown to be a good proxy for the original (deterministic) DNN both in terms of performance and the learned representations. We then develop a rigorous estimator for $I(X;T)$ in noisy DNNs and observe compression in various models. By relating $I(X;T)$ in the noisy DNN to an information-theoretic communication problem, we show that compression is driven by the progressive clustering of hidden representations of inputs from the same class. Several methods to directly monitor clustering of hidden representations, both in noisy and deterministic DNNs, are used to show that meaningful clusters form in the $T$ space. Finally, we return to the estimator of $I(X;T)$ employed in past works, and demonstrate that while it fails to capture the true (vacuous) mutual information, it does serve as a measure for clustering. This clarifies the past observations of compression and isolates the geometric clustering of hidden representations as the true phenomenon of interest.

## 1 INTRODUCTION

Recent work by Shwartz-Ziv & Tishby (2017) uses the Information Bottleneck framework (Tishby et al., 1999; Tishby & Zaslavsky, 2015) to study the dynamics of DNN learning. The framework considers the mutual information pair $\big(I(X;T_\ell), I(Y;T_\ell)\big)$ between the input $X$ or the label $Y$ and the network's hidden layers $T_\ell$. Plotting the evolution of these quantities during training, Shwartz-Ziv & Tishby (2017) made two interesting observations: (1) while $I(Y;T_\ell)$ remains mostly constant as the layer index $\ell$ increases, $I(X;T_\ell)$ decreases, suggesting that layers gradually shed irrelevant information about $X$; and (2) after an initial fitting phase, there is a long compression phase during which $I(X;T_\ell)$ slowly decreases. It was suggested that this compression is responsible for the generalization performance of DNNs. A follow-up paper (Saxe et al., 2018) contends that compression is not inherent to DNN training, claiming double-sided saturating nonlinearities yield compression while single-sided/non-saturating ones do not necessarily compress.

Shwartz-Ziv & Tishby (2017) and Saxe et al. (2018) present many plots of $\big(I(X;T_\ell), I(Y;T_\ell)\big)$ evolution across training epochs. These plots, however, are inadvertently misleading: they show a dynamically changing $I(X;T_\ell)$ when the true mutual information is provably either infinite or a constant independent of the DNN's parameters (see (Amjad & Geiger, 2018) for a discussion of further degeneracies related to to the Information Bottleneck framework). Recall that the mutual information $I(X;T_\ell)$ is a functional of the joint distribution of $(X,T_\ell) \sim P_{X,T_\ell} = P_X P_{T_\ell|X}$, and that, in standard DNNs, $T_\ell$ is a deterministic function of $X$. Hence, if $P_X$ is continuous, then so is $T_\ell$, and thus $I(X;T_\ell) = \infty$ (cf. (Polyanskiy & Wu, 2012-2017, Theorem 2.4)). If $P_X$ is discrete (e.g., when the features are discrete or if $X$ adheres to an empirical distribution over the dataset), then the mutual information is a finite constant that does not depend on the parameters of the DNN. Specifically, for deterministic DNNs, the mapping from a discrete $X$ to $T_\ell$ is injective for strictly

Figure 1: $I\big(X; \mathrm{Bin}(T_\ell)\big)$ vs. epochs for different bin sizes and the model in Shwartz-Ziv & Tishby (2017). The curves converge to $\ln(2^{12}) \approx 8.3$ for small bins, per the 12-bit uniformly distributed $X$.

monotone nonlinearities such as tanh or sigmoid, except for a measure-zero set of weights. In other words, deterministic DNNs can encode all information about a discrete $X$ in arbitrarily fine variations of $T_\ell$, causing no loss of information and implying $I(X; T_\ell) = H(X)$, even if deeper layers $\ell$ have fewer neurons.

The compression observed in Shwartz-Ziv & Tishby (2017) and Saxe et al. (2018) therefore cannot be due to changes in mutual information. This discrepancy between theory and experiments originates from a theoretically unjustified discretization of neuron values in their approximation of $I(X; T_\ell)$. To clarify, the quantity computed and plotted in these works is $I(X; \mathrm{Bin}(T_\ell))$, where $\mathrm{Bin}$ is a per-neuron discretization of each hidden activity of $T_\ell$ into a user-selected number of bins. This $I\big(X; \mathrm{Bin}(T_\ell)\big)$ is highly sensitive to the selection of bin size (as illustrated in Fig. 1) and does not track $I(X; T_\ell)$ for any choice of bin size.[1] Nonetheless, compression results based on $I\big(X; \mathrm{Bin}(T_\ell)\big)$ are observed by Shwartz-Ziv & Tishby (2017) and Saxe et al. (2018) in many interesting cases.

To understand this curious phenomenon we first develop a rigorous framework for tracking the flow of information in DNNs. In particular, to ensure $I(X; T_\ell)$ is meaningful for studying the learned representations, we need to make the map $X \mapsto T_\ell$ a stochastic parameterized channel whose parameters are the DNN's weights and biases. We identify several desirable criteria that such a stochastic DNN framework should fulfill for it to provide meaningful insights into commonly used practical systems. (1) The stochasticity should be intrinsic to the operation of the DNN, so that the characteristics of mutual information measures are related to the learned internal representations, and not to an arbitrary user-defined parameter. (2) The stochasticity should relate the mutual information to the deterministic binned version $I\big(X; \mathrm{Bin}(T_\ell)\big)$, since this is the object whose compression was observed; this requires the injected noise to be isotropic over the domain of $T_\ell$ analogously to the per-neuron binning operation. And most importantly, (3) the network trained under this stochastic model should be closely related to those trained in practice.

We propose a stochastic DNN framework in which independent and identically distributed (i.i.d.) Gaussian noise is added to the output of each of the DNN's neurons. This makes the map from $X$ to $T_\ell$ stochastic, ensures the data processing inequality (DPI) is satisfied, and makes $I(X; T_\ell)$ reflect the true operating conditions of the DNN, following Point (1). Since the noise is centered and isotropic, Point (2) holds. As for Point (3), Section 2 experimentally shows the DNN's learned representations and performance are not meaningfully affected by the addition of noise, for variances $\beta^2$ not too large. Furthermore, randomness during training has long been used to improve neural network performance, e.g., to escape poor local optima (Hinton et al., 1984), improve generalization performance (Srivastava et al., 2014), encourage learning of disentangled representations (Achille & Soatto, 2018), and ensure gradient flow with hard-saturating nonlinearities (Gulcehre et al., 2016).

Under the stochastic model, $I(X; T_\ell)$ has no exact analytic expression and is impossible to approximate numerically. In Section 3 we therefore propose a sampling technique that decomposes the estimation of $I(X; T_\ell)$ into several instances of a simpler differential entropy estimation problem: estimating $h(S + Z)$ given $n$ samples of the $d$-dimensional random vector $S$ and knowing the distribution of $Z \sim \mathcal{N}(0, \beta^2 \mathrm{I}_d)$. We analyze this problem theoretically and show that *any* differential entropy estimator over the noisy DNN requires at least exponentially many samples in the dimension $d$. Leveraging the explicit modeling of $S + Z$, we then propose a new estimator that converges

---

[1] Another approach taken in Saxe et al. (2018) considers $I(X; T_\ell + Z)$ (instead of $I\big(X; \mathrm{Bin}(T_\ell)\big)$), where $Z$ is an independent Gaussian with a user-defined variance. This approach has two issues: (i) the values as a function of $\ell$ may violate the data processing inequality, and (ii) they do not reflect the operation of the actual DNN, which was trained without noise. We focus on $I\big(X; \mathrm{Bin}(T_\ell)\big)$ because it was commonly used in Shwartz-Ziv & Tishby (2017) and Saxe et al. (2018), and since both methods have a similar effect of blurring $T_\ell$.

as $O\big((\log n)^{d/4}/\sqrt{n}\big)$, which significantly outperforms the convergence rate of general-purpose differential entropy estimators when applied to the noisy DNN framework.

We find that $I(X;T_\ell)$ exhibits compression in many cases during training of small DNN classifiers. To explain compression in an insightful yet rigorous manner, Section 4 relates $I(X;T_\ell)$ to the well-understood notion of data transmission over additive white Gaussian noise (AWGN) channels. Namely, $I(X;T_\ell)$ is the aggregate information transmitted over the channel $P_{T_\ell|X}$ with input $X$ drawn from a constellation defined by the data samples and the noisy DNN parameters. As training progresses, the representations of inputs from the same class tend to cluster together and become increasingly indistinguishable at the channel's output, thereby decreasing $I(X;T_\ell)$. Furthermore, these clusters tighten as one moves into deeper layers, providing evidence that the DNN's layered structure progressively improves the representation of $X$ to increase its relevance for $Y$.

Finally, we examine clustering in deterministic DNNs. We identify methods for measuring clustering that are valid for both noisy and deterministic DNNs, and show that clusters of inputs in learned representations typically form in both cases. We complete the circle back to $I\big(X;\mathsf{Bin}(T_\ell)\big)$ by clarifying why this binned mutual information measures clustering. This explains what previous works were actually observing: not compression of mutual information, but increased clustering by hidden representations. The geometric clustering of hidden representations is thus the fundamental phenomenon of interest, and we aim to test its connection to generalization performance, theoretically and experimentally, in future work.

## 2 PRELIMINARY DEFINITIONS

**Noisy DNNs:** For integers $k \le \ell$, let $[k:\ell] \triangleq \big\{i \in \mathbb{Z} \big| k \le i \le \ell\big\}$ and use $[\ell]$ when $k = 1$. Consider a noisy DNN with $L+1$ layers $\{T_\ell\}_{\ell \in [0:L]}$, with input $T_0 = X$ and output $T_L$. The $\ell$th hidden layer, $\ell \in [L-1]$, is described by $T_\ell = f_\ell(T_{\ell-1}) + Z_\ell$, where $f_\ell : \mathbb{R}^{d_{\ell-1}} \to \mathbb{R}^{d_\ell}$ is a deterministic function of the previous layer and $Z_\ell \sim \mathcal{N}\big(0, \beta^2 \mathrm{I}_{d_\ell}\big)$; no noise is injected to the output, i.e., $T_L = f_L(T_{L-1})$. We set $S_\ell \triangleq f_\ell(T_{\ell-1})$ and use $\varphi$ for the probability density function (PDF) of $Z_\ell$. The functions $\{f_\ell\}_{\ell \in [L]}$ can represent any type of layer (fully connected, convolutional, max-pooling, etc.). Fig. 2 shows a neuron in the $\ell$th layer of a noisy DNN.

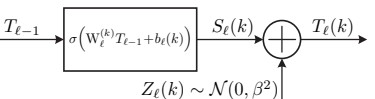

Figure 2: $k$th noisy neuron in layer $\ell$ with nonlinearity $\sigma$; $\mathrm{W}_\ell^{(k)}$ and $b_\ell(k)$ are the $k$th row/entry of the weight matrix and the bias, respectively.

To explore the relation between noisy and deterministic DNNs under conditions representative of current machine learning practices, we trained four-layer convolutional neural networks (CNNs) on MNIST (LeCun et al., 1999). The CNNs used different levels of internal noise, including no noise, and one used dropout in place of additive noise. We measured their performance on the validation set and characterized the cosine similarities between their internal representations. Full details of the CNN architecture and training procedure are in Supplement 9.3. The results in Table 1 show small amounts of internal additive noise ($\beta \le 0.1$) have a minimal impact on classification performance, while dropout strongly improves it. The histograms in Fig. 3 show that the noisy (for small $\beta$) and dropout models learn internal representations similar to

| Model | # Errors |
|---|---|
| Deterministic | $50 \pm 4.6$ |
| Noisy ($\beta = 0.05$) | $50 \pm 5.0$ |
| Noisy ($\beta = 0.1$) | $51 \pm 6.9$ |
| Noisy ($\beta = 0.2$) | $86 \pm 9.8$ |
| Noisy ($\beta = 0.5$) | $2200 \pm 520$ |
| Dropout ($p = 0.2$) | $39 \pm 3.9$ |

Table 1: Total MNIST validation errors for different models, showing mean $\pm$ standard deviation over eight initial random seeds.

the representations learned by the deterministic model. In this high-dimensional space, unrelated representations would create cosine similarity histograms with zero mean and standard deviation between 0.02–0.3, so the observed values are quite large. As expected, dissimilarity increases as the noise increases, and similarity is lower for the internal layers (2 and 3).

**Mutual Information:** Noisy DNNs induce a stochastic map from $X$ to the rest of the network, described by the conditional distribution $P_{T_1,\dots,T_L|X}$. The corresponding PDF[2] is $p_{T_1,\dots,T_L|X=x}$. Its marginals are denoted by keeping only the relevant variables in the subscript. Let $\mathcal{X} \triangleq \{x_i\}_{i \in [m]}$ be

---

[2]$P_{T_1,\dots,T_L|X=x}$ is absolutely continuous with respect to (w.r.t.) the Lebesgue measure for all $x \in \mathcal{X}$.

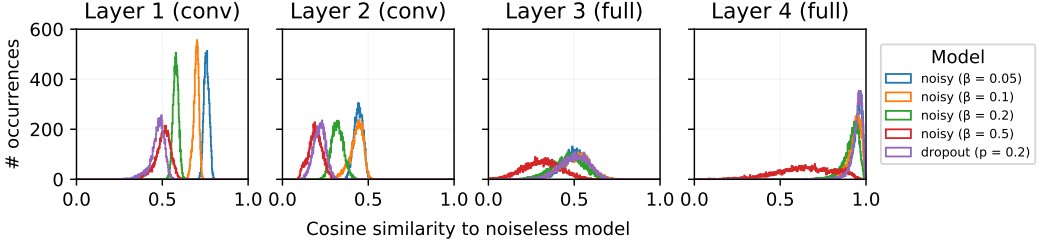

Figure 3: Histograms of cosine similarities between internal representations of deterministic, noisy, and dropout MNIST CNN models. To encourage comparable internal representations, all models were initialized with the same random weights and accessed the training data in the same order.

the input dataset, and $\hat{P}_{\mathcal{X}}$ be its empirical distribution, described by the probability mass function (PMF) $\hat{p}_{\mathcal{X}}(x) = \frac{1}{m}\sum_{i \in [m]} \mathbb{1}_{\{x_i = x\}}$, for $x \in \mathcal{X}$. Since data sets typically contain no repetitions, we assume $\hat{p}_{\mathcal{X}}(x) = \frac{1}{m}, \forall x \in \mathcal{X}$. The input and the hidden layers are jointly distributed according to[3] $P_{X,T_1,\ldots,T_L} \triangleq \hat{P}_{\mathcal{X}} P_{T_1,\ldots,T_L|X}$, under which $X - T_1 - \ldots - T_{L-1} - T_L$ forms a Markov chain. For each $\ell \in [L-1]$, we study the mutual information (Supplement 7 explains this factorization)

$$I(X; T_\ell) \triangleq \int_{\mathcal{X} \times \mathbb{R}^{d_\ell}} \mathrm{d}P_{X,T_\ell} \log\left(\frac{\mathrm{d}P_{X,T_\ell}}{\mathrm{d}P_X \times P_{T_\ell}}\right) = h(p_{T_\ell}) - \frac{1}{m}\sum_{i \in [m]} h(p_{T_\ell|X=x_i}), \qquad (1)$$

where $\log(\cdot)$ is with respect to the natural base. Although $P_{T_\ell}$ and $P_{T_\ell|X}$ are readily sampled from using the DNN's forward pass, these distributions are too complicated (due to the composition of Gaussian noises and nonlinearities) to analytically compute $I(X; T_\ell)$ or even to evaluate their densities at the sampled points. Therefore, we must estimate $I(X; T_\ell)$ directly from the available samples.

## 3  MUTUAL INFORMATION ESTIMATION OVER NOISY DNNS

Expanding $I(X; T_\ell)$ as in (1), our goal is to estimate $h(p_{T_\ell})$ and $h(p_{T_\ell|X=x}), \forall x \in \mathcal{X}$: a problem that we show is hard in high dimensions. Each differential entropy term is estimated and computed via a two-step process. First, we develop the *sample propagation* (SP) estimator, which exploits the ability to propagate samples up the DNN layers and the known noise distribution. This estimator approximates each true entropy by the differential entropy of a *known* Gaussian mixture (defined only through the available resources: the samples we obtain from the DNN and the noise parameter). This estimate is shown to converge to the true entropy when the number of samples grows. However, since the entropy of a Gaussian mixture has no closed-form expression, in the second (computational) step we use Monte Carlo (MC) integration to numerically evaluate it.

### 3.1  THE SAMPLE-PROPAGATION DIFFERENTIAL ENTROPY ESTIMATOR

In what follows, we denote the empirical PMF associated with a set $\mathcal{A} = \{a_i\}_{i \in [n]} \subset \mathbb{R}^d$ by $\hat{p}_{\mathcal{A}}$.

**Unconditional Entropy:** Since $T_\ell = S_\ell + Z_\ell$, where $S_\ell$ and $Z_\ell$ are independent, we have $p_{T_\ell} = p_{S_\ell} * \varphi$. To estimate $h(p_{T_\ell})$, let $\{\hat{x}_j\}_{j \in [n]}$ be $n$ i.i.d. samples from $P_X$. Feed each $\hat{x}_j$ into the DNN and collect the outputs it produces at the $(\ell - 1)$-th layer. The function $f_\ell$ is then applied on each collected output to obtain $\mathcal{S}_\ell \triangleq \{s_{\ell,1}, s_{\ell,2}, \ldots, s_{\ell,n}\}$, which is a set of $n$ i.i.d. samples from $p_{S_\ell}$. We estimate $h(p_{T_\ell})$ by $h(\hat{p}_{\mathcal{S}_\ell} * \varphi)$, which is the differential entropy of a Gaussian mixture with centers $s_{\ell,j}, j \in [n]$. The term $h(\hat{p}_{\mathcal{S}_\ell} * \varphi)$ is referred to as the SP estimator of $h(p_{T_\ell}) = h(p_{S_\ell} * \varphi)$.

**Conditional Entropies:** Fix $i \in [m]$ and consider the estimation of $h(p_{T_\ell|X=x_i})$. Note that $p_{T_\ell|X=x_i} = p_{S_\ell|X=x_i} * \varphi$ since $Z_\ell$ is independent of $(X, T_{\ell-1})$. To sample from $p_{S_\ell|X=x_i}$, we feed $x_i$ into the DNN $n_i$ times, collect outputs from $T_{\ell-1}$ corresponding to different noise realizations, and apply $f_\ell$ on each. The obtained samples $\mathcal{S}_\ell^{(i)} \triangleq \{s_{\ell,1}^{(i)}, s_{\ell,2}^{(i)}, \ldots, s_{\ell,n_i}^{(i)}\}$ are i.i.d. according to $p_{S_\ell|X=x_i}$. Each $h(p_{T_\ell|X=x_i}) = h(p_{S_\ell|X=x_i} * \varphi)$ is estimated by the SP estimator $h(\hat{p}_{\mathcal{S}_\ell^{(i)}} * \varphi)$.[4]

**Mutual Information Estimator:** Combining the above described pieces, we estimate $I(X; T_\ell)$ by

$$\widehat{I(X; T_\ell)} = h(\hat{p}_{\mathcal{S}_\ell} * \varphi) - \frac{1}{m}\sum_{i \in [m]} h\left(\hat{p}_{\mathcal{S}_\ell^{(i)}} * \varphi\right). \qquad (2)$$

---

[3] We set $X \sim \mathsf{Unif}(\mathcal{X})$ to conform with past works (Shwartz-Ziv & Tishby, 2017; Saxe et al., 2018).
[4] For $\ell = 1$, we have $h(T_1|X) = h(Z_1) = \frac{d_1}{2}\log(2\pi e\beta^2)$ because its previous layer is $X$ (fixed).

## 3.2 Theoretical Guarantees and Computing the Estimator

The above sampling procedure unifies the estimation of $h(p_{T_\ell})$ and $\{h(p_{T_\ell|X=x})\}_{x\in\mathcal{X}}$ into a single new differential entropy estimation problem: estimate $h(p_S * \varphi)$ based on i.i.d. samples $S^n \triangleq (S_i)_{i\in[n]}$ from $p_S$ and knowledge of $\varphi$. The SP estimator solution approximates $h(p_S * \varphi)$ by $\hat{h}_{\mathsf{SP}}(S^n, \varphi) \triangleq h(\hat{p}_{S^n} * \varphi)$, where $\hat{p}_{S^n}$ is the empirical PMF induced by $S^n$. Before analyzing the performance of $\hat{h}_{\mathsf{SP}}$, we note that this estimation problem is statistically difficult in the sense that any good estimator of $h(p_S * \varphi)$ based on $S^n$ and $\varphi$ requires exponentially many samples in $d$ (Theorem 2 from Supplement 10). Nonetheless, the following theorem shows that the SP estimator absolute-error risk converges at a satisfactory rate (Theorem 4 from Supplement 10 states this with all constants explicit, and Theorem 5 gives the results for ReLU).

**Theorem 1** *Fix $\beta > 0$, $d \geq 1$, and let $\mathcal{F}_d$ be the class of d-dimensional PDFs supported inside $[-1,1]^d$. We have:* $\sup_{p_S\in\mathcal{F}_d} \mathbb{E}\big|h(p_S * \varphi) - \hat{h}_{\mathsf{SP}}(S^n, \varphi)\big| = O\big((\log n)^{d/4}/\sqrt{n}\big)$.

Evaluating the SP estimator $\hat{h}_{\mathsf{SP}}(S^n, \varphi)$ of the true entropy $h(p_S * \varphi)$ requires computing the differential entropy of the (known) Gaussian mixture $\hat{p}_{s^n} * \varphi$ since

$$\hat{h}_{\mathsf{SP}}(S^n, \varphi) = h\big(\hat{p}_{s^n} * \varphi\big). \tag{3}$$

Noting that the differential entropy $h(p) = -\mathbb{E}_{X\sim p}[\log p(X)]$, we rewrite the SP estimator as

$$\hat{h}_{\mathsf{SP}}(S^n, \varphi) = h(G) = -\mathbb{E}\Big[\log\big((\hat{p}_{s^n} * \varphi)(G)\big)\Big], \tag{4}$$

where $G \sim \hat{p}_{s^n} * \varphi$ is distributed according to the Gaussian mixture.

We numerically approximate the right-hand side of (4) via efficient Monte Carlo (MC) integration (Robert, 2004). Specifically, we generate $n_{\mathsf{MC}}$ i.i.d. samples from $\hat{p}_{s^n} * \varphi$ and approximate the expectation by an empirical average. This unbiased approximation achieves a mean squared error of $O\big((n \cdot n_{\mathsf{MC}})^{-1}\big)$ (Supplement 10). This approximation thus only adds a negligible amount to the error of the SP estimator $\big|h(p_S * \varphi) - \hat{h}_{\mathsf{SP}}(S^n, \varphi)\big|$ itself. There are other ways to numerically evaluate this expectation, such as the Gaussian mixture bounds from Kolchinsky & Tracey (2017); however, our proposed method is the fastest approach of which we are aware.

**Remark 1 (Choosing Noise Parameter and Number of Samples)** *We describe practical guidelines for selecting the noise standard deviation $\beta$ and the number of samples $n$ for estimating $I(X; T_\ell)$ in an actual classifier. Ideally, $\beta$ should be treated as a hyperparameter tuned to optimize the performance of the classifier on held-out data, since internal noise serves as a regularizer similar to dropout. In practice, we find it is sometimes necessary to back off from the $\beta$ value that optimizes performance to a higher value to ensure accurate estimation of mutual information (the smaller $\beta$ is, the more samples our estimator requires), depending on factors such as the dimensionality of the layer being analyzed and the number of data samples available for a task.*

*The number of samples $n$ can be selected using the bound in Theorem 1, but because this theorem is a worst-case result, in practice it is quite pessimistic. Specifically, generating the estimated mutual information curves shown in Section 5 requires running the SP estimator multiple times[5], which makes the number of samples dictated by Theorem 1 infeasible. To overcome this computational burden while adhering to the theoretical result, we tested the value of $n$ given by the theorem on a few points of each curve and reduced it until the overall computation cost became reasonable. To ensure estimation accuracy was not compromised we empirically tested that the estimate remained stable.*

*As a concrete example, to achieve an error bound of 5% of Fig. 5 plot's vertical scale (which amounts to an $0.4$ absolute error bound), the number of samples required by Theorem 1 is $n = 4 \cdot 10^9$. This number is too large for our computational budget. Performing the above procedure for reducing $n$, we find good accuracy is achieved for $n = 4 \cdot 10^6$ samples (Theorem 1 has the pessimistic error bound of $3.74$ for this value). Adding more samples beyond this value does not change the results.*

---

[5]Each $I(X; T_\ell)$, for a given set of DNN parameters, involves computing $m+1$ differential entropy estimates, and our experiments estimate the trajectory of $I(X; T_\ell)$ across training epochs.

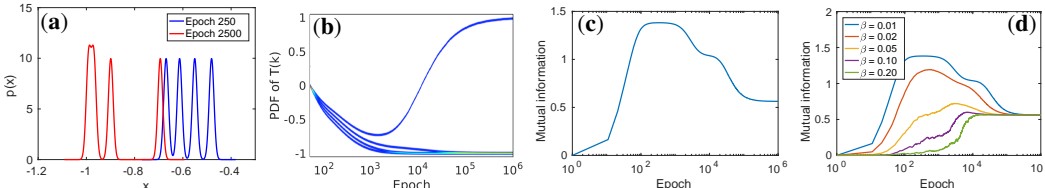

Figure 4: Single-layer tanh network: (a) the density $p_{T(k)}$ at epochs $k = 250, 2500$; (b) $p_{T(k)}$ and (c) $I\big(X; T(k)\big)$ as a function of $k$; and (d) mutual information as a function of weight $w$ with bias $-2w$.

## 4 COMPRESSION AND CLUSTERING: A MINIMAL EXAMPLE

Before presenting our empirical results, we connect compression to clustering using an information-theoretic perspective. Consider a single noisy neuron with a one-dimensional input $X$. Let $T(k) = S(k) + Z$ be the neuron's output at epoch $k$, where $S(k) \triangleq \sigma(w_k X + b_k)$, for a strictly monotone nonlinearity $\sigma$, and $Z \sim \mathcal{N}(0, \beta^2)$. Invariance of mutual information to invertible operations implies

$$I\big(X; T(k)\big) = I\big(\sigma(w_k X + b_k); \sigma(w_k X + b_k) + Z\big) = I\big(S(k); S(k) + Z\big). \tag{5}$$

From an information-theoretic perspective, $I\big(S(k); S(k) + Z\big)$ is the aggregate information transmitted over an AWGN channel with input constellation $\mathcal{S}_k \triangleq \big\{\sigma(w_k x + b_k) \mid x \in \mathcal{X}\big\}$. In other words, $I\big(S(k); S(k) + Z\big)$ is a measure of how distinguishable the symbols of $\mathcal{S}_k$ are when composed with Gaussian noise (roughly equals $\log$ of the number of resolvable clusters under noise level $\beta$). Since the distribution of $T(k) = S(k) + Z$ is a Gaussian mixture with means $s \in \mathcal{S}_k$, the closer two constellation points $s$ and $s'$ are, the more overlapping the Gaussians around them will be. Hence reducing point spacing in $\mathcal{S}_k$ (by changing $w_k$ and $b_k$) directly reduces $I\big(X; T(k)\big)$.

Let $\sigma = \tanh$ and $\beta = 0.01$, and set $\mathcal{X} = \mathcal{X}_{-1} \cup \mathcal{X}_1$, with $\mathcal{X}_{-1} = \{-3, -1, 1\}$ and $\mathcal{X}_1 = \{3\}$, labeled $-1$ and $1$, respectively. We train the neuron using mean squared loss and gradient descent (GD) with a fixed learning rate of $0.01$ to best illustrate the behavior of $I\big(X; T(k)\big)$. The Gaussian mixture $p_{T(k)}$ is plotted across epochs $k$ in Fig. 4(a)-(b). The learned bias is approximately $-2.3w$, ensuring that the tanh transition region correctly divides the two classes. Initially $w = 0$, so all four Gaussians in $p_{T(0)}$ are superimposed. As $k$ increases, the Gaussians initially diverge, with the three from $\mathcal{X}_{-1}$ eventually re-converging as they each meet the tanh boundary. This is reflected in the mutual information trend in Fig. 4(c), with the dips in $I\big(X; T(k)\big)$ around $k = 10^3$ and $k = 10^4$ corresponding to the second and third Gaussians respectively merging into the first. Thus, there is a direct connection between clustering and compression. Fig. 4(d) shows the mutual information for different noise levels $\beta$ as a function of epoch. For small $\beta$ (as above) the $\mathcal{X}_{-1}$ Gaussians are distinct and merge in two stages as $w$ grows. For larger $\beta$, however, the $\mathcal{X}_{-1}$ Gaussians are indistinguishable for any $w$, making $I(X; T)$ only increase as the two classes gradually separate. A similar example for a two-neuron network with leaky-ReLU nonlinearities is provided in the Supplement 8.

## 5 EMPIRICAL RESULTS

We now show the observations from our minimal examples also hold for two larger networks. Namely, the presented experiments demonstrate the compression of mutual information in noisy networks is driven by clustering of internal representation, and that deterministic networks cluster samples as well (despite $I(X; T_\ell)$ being constant over these systems). The DNNs we consider are: (1) the small, fully connected network (FCN) studied in (Shwartz-Ziv & Tishby, 2017; Saxe et al., 2018), which we call the *SZT model*; and (2) a convolutional network for MNIST classification, called *MNIST CNN*. We present selected results; additional details and experiments are found in the supplement.

**SZT model:**

Consider the data and model of Shwartz-Ziv & Tishby (2017) for binary classification of 12-dimensional inputs using a fully connected 12–10–7–5–4–3–2 architecture. The FCN was tested with tanh and ReLU nonlinearities as well as a linear model. Fig. 5(a) presents results for the SZT model with tanh nonlinearity and $\beta = 0.005$ (test classification accuracy 99%), showing the relationship across training epochs between estimated $I(X; T_\ell)$, train/test losses and the distribution of neuron values in 5 layers (layers 0 ($d_0 = 12$) and 6 ($d_7 = 2$) are not shown). The rise and fall of mutual information corresponds to how spread out or clustered the representation in each layer are. For

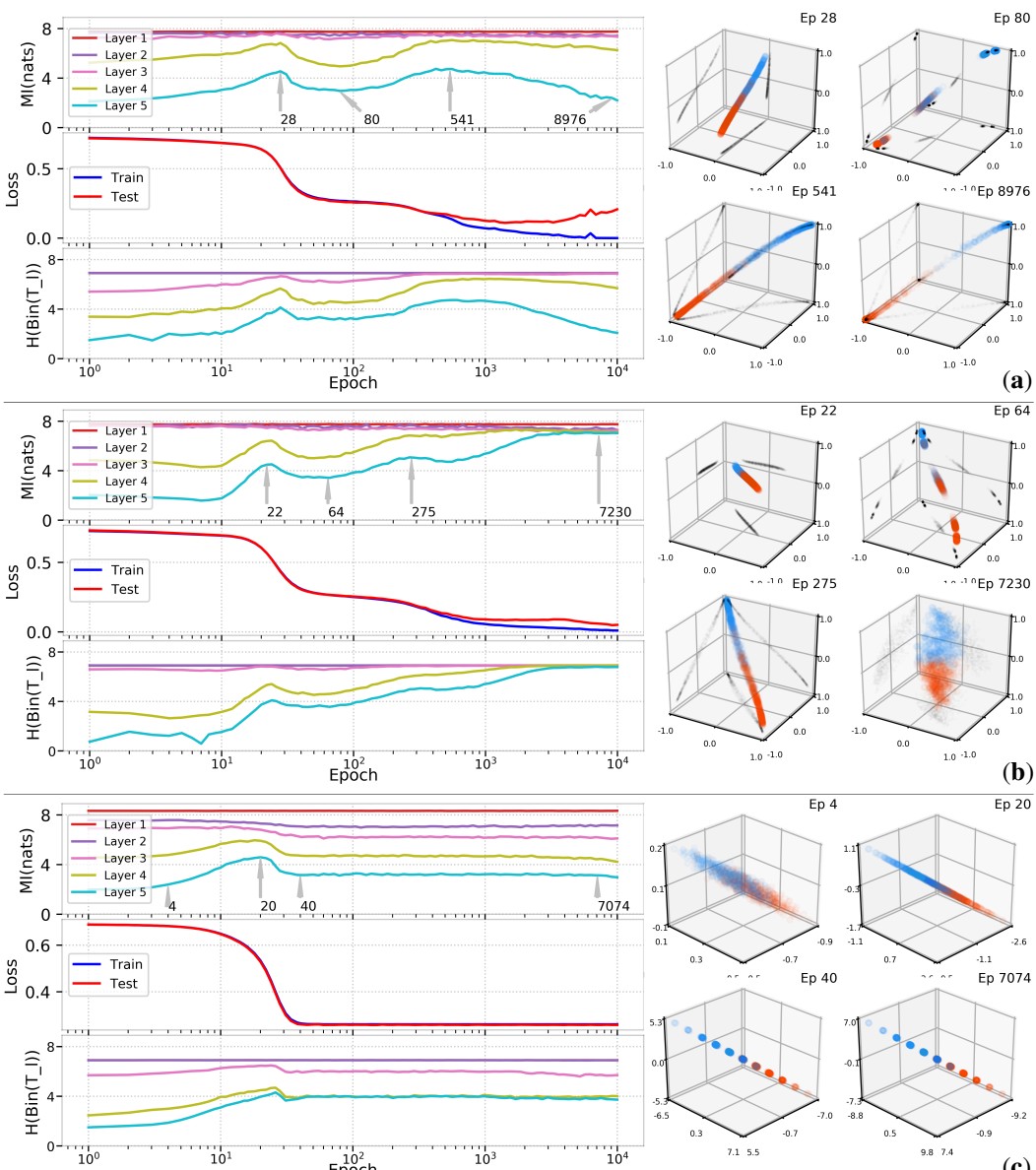

Figure 5: (a) Evolution of $I(X; T_\ell)$ and training/test losses across training epochs for the SZT model with $\beta = 0.005$ and tanh nonlinearities. The scatter plots show the values of Layer 5 ($d_5 = 3$) at the arrow-marked epochs on the mutual information plot. The bottom plot shows $H(\text{Bin}(T_\ell))$ across epochs for bin size $B = 10\beta$. (b) Same setup as in (a) but with regularization that encourages orthonormal weight matrices. (c) SZT model with $\beta = 0.01$ and *linear* activations.

example, $I(X; T_5)$ grows until epoch 28, when the Gaussians move away from each other along a curve (see scatter plots on the right). Around epoch 80 they start clustering and $I(X; T_5)$ drops. At the end of training, the saturating tanh nonlinearities push the Gaussians to two furthest corners of the cube, reducing $I(X; T_5)$ even more.

To confirm that clustering (via saturation) was central to the compression observed in Fig. 5(a), we also trained the model using the regularization from (Cisse et al., 2017) (test classification accuracy 96%), which encourages orthonormal weight matrices. The results are shown in Fig. 5(b). Apart from minor initial fluctuations, the bulk of compression is gone. The scatter plots show that the vast majority of neurons do not saturate and no clustering is observed at the later stages of training. Saturation is not the only mechanism that can cause clustering and consequently reduce $I(X; T_\ell)$. For example, in Fig. 5(c) we illustrate the clustering behavior in a linear SZT model (test

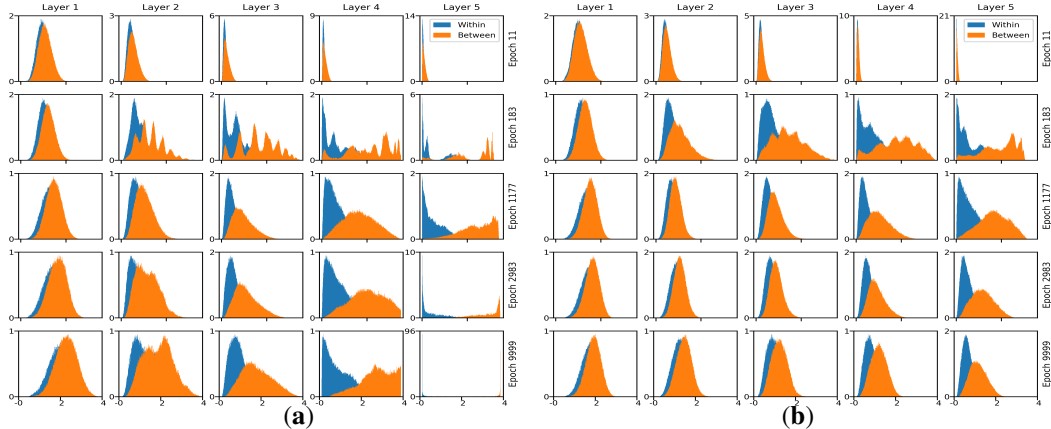

Figure 6: (a) Histogram of within- and between-class pairwise distances for SZT model with tanh non-linearities and additive noise $\beta = 0.005$. (b) Same as (a) but training with weight normalization.

classification accuracy 89%). As seen from the scatter plots, due to the formation of several clusters and projection to a lower dimensional space, $I(X; T_\ell)$ drops even without the nonlinearities. The results in Fig. 5(a) and (b) also show that the relationship between compression and generalization performance is not a simple one. In Fig. 5(a), the test loss begins to increase at roughly epoch 3200 and continues to increase until training ends, while at the same time compression occurs in layers 4 and 5. In contrast, in Fig. 5(b) the test loss does not increase, and compression does not occur in layers 4 and 5. We believe that this is a subject that deserves further examination in future work.

To provide another perspective on clustering that is sensitive to class membership, we compute histograms of pairwise distances between representations of samples, distinguishing within-class distances from between-class distances. Fig. 6 shows histograms for the SZT models from Figs. 5(a) and (b). As training progresses, the formation of clusters is clearly seen (layer 3 and beyond) for the unnormalized SZT model in Fig. 5(a). In the normalized model (Fig. 5(b)), however, no tight clustering is apparent, supporting the connection between clustering and compression.

Once clustering is identified as the source of compression, we focus on it as the point of interest. To measure clustering, the discrete entropy of $\mathrm{Bin}(T_\ell)$ is considered, where the number of equal-sized bins, $B$, is a tuning parameter. Note that $\mathrm{Bin}(T_\ell)$ partitions the dynamic range (e.g., $[-1, 1]^{d_\ell}$ for a tanh layer) into $B^{d_\ell}$ cells or bins. When hidden representations are spread out, many bins will be non-empty, each assigned with a positive probability mass. On the other hand, for clustered representations, the distribution is concentrated on a small number of bins, each with relatively high probability. Recalling that discrete entropy is maximized by the uniform distribution, we see why reduction in $H\big(\mathrm{Bin}(T_\ell)\big)$ measures clustering.

To illustrate this measure, we compute $H\big(\mathrm{Bin}(T_\ell)\big)$ for each of the SZT models using bin size $B = 10\beta$ (bottom plots in Fig. 5(a), (b) and (c)). We can see a clear correspondence between $H\big(\mathrm{Bin}(T_\ell)\big)$ and $I(X; T_\ell)$, indicating that although $H\big(\mathrm{Bin}(T_\ell)\big)$ does not capture the exact value of $I(X; T_\ell)$, it follows this mutual information in measuring clustering. This is particularly important when moving back to deterministic DNNs, where $I(X; T_\ell)$ is no longer an informative measure, being either a constant or infinity, for discrete or continuous $X$, respectively.

Fig. 1 shows $H\big(\mathrm{Bin}(T_\ell)\big)$ for the deterministic SZT model ($\beta = 0$). The bin size is a free parameter, and depending on its value, $H\big(\mathrm{Bin}(T_\ell)\big)$ reveals different clustering granularities. Moreover, since in deterministic networks $T_\ell = f_\ell(X)$, for a deterministic map $f_\ell$, we have $H\big(\mathrm{Bin}(T_\ell)\big|X\big) = 0$, and therefore $I\big(X; \mathrm{Bin}(T_\ell)\big) = H\big(\mathrm{Bin}(T_\ell)\big)$. Thus, the plots from (Shwartz-Ziv & Tishby, 2017), (Saxe et al., 2018) and our Figs. 1 and 5(a), (b) and (c) all show the entropy of the binned $T_\ell$.

**MNIST CNN:** We now examine a model that is more representative of current machine learning practice: the MNIST CNN trained with dropout from Section 2. Fig. 7 portrays the near-injective behavior of this model. Even when only two bins are used to compute $H\big(\mathrm{Bin}(T_\ell)\big)$, it takes values that are approximately $\ln(10000) = 9.210$, for all layers and training epochs, even though the two convolutional layers use max-pooling.

This binning merges two samples in the validation set, so the input has $H\big(\text{Bin}(T_\ell)\big) = 9.209$. While Fig. 7 does not show compression at the level of entire layers, computing $H\big(\text{Bin}(T_\ell(k))\big)$ for individual units $k$ in layer 3 reveals a gradual decrease over epochs 1–128. To quantify this trend, we computed linear regressions predicting $H\big(\text{Bin}(T_\ell(k))\big)$ from the epoch index, for all units $k$ in layer 3. Then we found the mean and standard deviation of the slope of the linear predictions. If most slopes are negative, then compression occurs during training at the level of individual units. For a range of bin sizes from $10^{-4}$–$10^{-1}$ the least negative mean slope was $-0.002$ nats/epoch with a maximum standard deviation of $0.001$, showing that most units undergo compression.

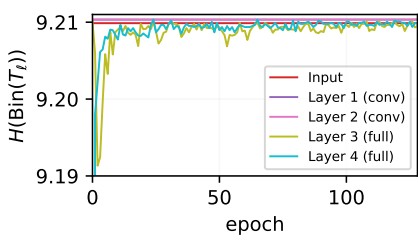

Figure 7: $H\big(\text{Bin}(T_\ell)\big)$ for the MNIST CNN, computed using two bins: $[-1, 0]$ and $(0, 1]$. The tiny range of the y axis shows the near injectivity of the model.

In Fig. 8 we show histograms of pairwise distances between MNIST validation set samples in the input (pixel) space and in the four layers of the CNN. The histograms were computed for epochs 0, 1, 32, and 128, where epoch 0 is the initial random weights and epoch 128 is the final weights. The histogram for the input shows that the mode of within-class pairwise distances is lower than the mode of between-class pairwise distances, but that there is substantial overlap. Layers 1 and 2, which are convolutional and therefore do not contain any units that receive the full input, do little to reduce this overlap, suggesting that the features learned in these layers are somewhat generic. In contrast, even after one epoch of training, layers 3 and 4, which are fully connected, separate the distribution of within-class distances from the distribution of between-class distances.

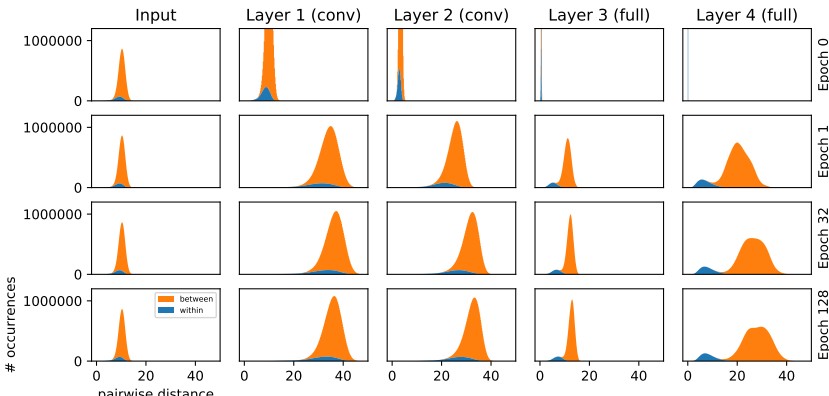

Figure 8: Histograms of within-class and between-class pairwise distances from the MNIST CNN.

To summarize, we made the following observations in our experiments. (i) Compression can be observed in a noisy network that is similar to the deterministic network in which (Shwartz-Ziv & Tishby, 2017) reported compression (upper left plot in Fig. 5(a)). (ii) Compression is caused by clustering of samples, with clusters most often comprising samples having the same class label, as seen in the scatter plots on the right sides of Figs. 5(a) and (c) and the distributions of pairwise distances between samples shown in Figs. 6 and 8. (iii) Regularization that limits the ability of a network to drive hidden units into saturation may limit or eliminate compression (and clustering) as seen in Fig. 5(b). Fig. 5 also demonstrated that $I(X; T_\ell)$ and $H\big(\text{Bin}(T_\ell)\big)$ are highly correlated, establishing the latter as an additional measure for clustering (applicable both in noisy and deterministic DNNs). (iv) Clustering of internal representations can also be observed in a somewhat larger, convolutional network trained on MNIST. While Fig. 7 shows that due to the dimensionality, $H\big(\text{Bin}(T_\ell)\big)$ fails to track compression in the larger CNN, strong evidence for clustering is found via estimates done at the level of individual units (described in the text on the MNIST CNN) and the analysis of pairwise distances between samples shown in Fig. 8.

## 6    CONCLUSIONS

In this work we reexamined the compression aspect of the Information Bottleneck theory (Shwartz-Ziv & Tishby, 2017), noting that fluctuations of $I(X; T_\ell)$ in deterministic networks with strictly

monotone nonlinearities are theoretically impossible. Setting out to discover the source of compression observed in past works, we: (i) created a rigorous framework for studying and accurately estimating information-theoretic quantities in DNNs whose weights are fixed; (ii) identified clustering of the learned representations as the phenomenon underlying compression; and (iii) demonstrated that the compression-related experiments from past works were in fact measuring this clustering through the lens of the binned mutual information. In the end, although binning-based measures do not accurately estimate mutual information, they are simple to compute and prove useful for tracking changes in clustering, which is the true effect of interest in deterministic DNNs. We believe that further study of geometric phenomena driven by DNN training is warranted to better understand the learned representations and to potentially establish connections with generalization.

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

# SUPPLEMENT TO ESTIMATING INFORMATION FLOW IN DNNS

**Anonymous authors**

## 7 MUTUAL INFORMATION

Let $(A, B)$ be a pair of random variables with values in the product set $\mathcal{A} \times \mathcal{B}$ and a joint distribution $P_{A,B}$ (whose marginals are denoted by $P_A$ and $P_B$). The mutual information between $A$ and $B$ is:

$$I(A; B) \triangleq \int_{\mathcal{A} \times \mathcal{B}} \mathrm{d}P_{A,B} \log \left( \frac{\mathrm{d}P_{A,B}}{\mathrm{d}P_A \times P_B} \right), \tag{5}$$

where $\frac{\mathrm{d}P_{A,B}}{\mathrm{d}P_A \times P_B}$ is the Radon-Nikodym derivative of $P_{A,B}$ with respect to the product measure $P_A \times P_B$. We are mostly interested in the scenario where $A$ is discrete with a probability mass function (PMF) $p_A$, and given $A = a \in \mathcal{A}$, $B$ is continuous with probability density function (PDF) $p_{B|A=a} \triangleq p_{B|A}(\cdot|a)$. In this case, (5) simplifies to

$$I(A; B) = \sum_{a \in \mathcal{A}} p_A(a) \int_{\mathcal{B}} p_{B|A}(b|a) \log \left( \frac{p_{B|A}(b|a)}{p_B(b)} \right) db. \tag{6}$$

Defining the differential entropy of a continuous random variable $C$ with PDF $p_C$ supported in $\mathcal{C}$ as[1]

$$h(C) = h(p_C) = - \int_{\mathcal{C}} p_C(c) \log p_C(c) dc, \tag{7}$$

the mutual information from (6) can also be expressed as

$$I(A; B) = h(p_B) - \sum_{a \in \mathcal{A}} p_A(a) h(p_{B|A=a}). \tag{8}$$

The subtracted term above is the *conditional differential entropy* of $B$ given $A$, denoted by $h(B|A)$.

## 8 TWO-NEURON LEAKY-RELU NETWORK EXAMPLE

To expand upon Section 4, we provide here a second example to illustrate the relation between clustering and compression of mutual information. In particular, this example also shows that as opposed to the claim from (Saxe et al., 2018), non-saturating nonlinearities can achieve compression. Consider the non-saturating Leaky-ReLU nonlinearity $R(x) \triangleq \max(x, x/10)$. Let $\mathcal{X} = \mathcal{X}_0 \cup \mathcal{X}_{1/4}$, with $\mathcal{X}_0 = \{1, 2, 3, 4\}$ and $\mathcal{X}_{1/4} = \{5, 6, 7, 8\}$, and labels $0$ and $1/4$, respectively. We train the network via GD with learning rate 0.001 and mean squared loss. Initialization (shown in Fig. 9(a)) was chosen to best illustrate the connection between the Gaussians' motion and mutual information. The network converges to a solution where $w_1 < 0$ and $b_1$ is such that the elements in $\mathcal{X}_{1/4}$ cluster. The output of the first layer is then negated using $w_2 < 0$ and the bias ensures that the elements in $\mathcal{X}_0$ are clustered without spreading out the elements in $\mathcal{X}_{1/4}$. Figs. 9(b) show the Gaussian motion at the output of the first layer and the resulting clustering. For the second layer (Fig. 9(c)), the clustered bundle $\mathcal{X}_{1/4}$ is gradually raised by growing $b_2$, such that its elements successively split as they cross the origin; further tightening of the bundle is due to shrinking $|w_2|$. Fig. 9(d) shows the mutual information of the first (blue) and second (red) layers. The merging of the elements in $\mathcal{X}_{1/4}$ after their initial divergence is clearly reflected in the mutual information. Likewise, the spreading of the bundle, and successive splitting and coalescing of the elements in $\mathcal{X}_{1/4}$ are visible in the spikes in the red mutual information curve. The figure also shows how the bounds on $I(X; T(k))$ precisely track its evolution.

---

[1]Throughout this work we interchanging use $h(C)$ and $h(p_C)$ for the differential entropy of $C \sim p_C$.

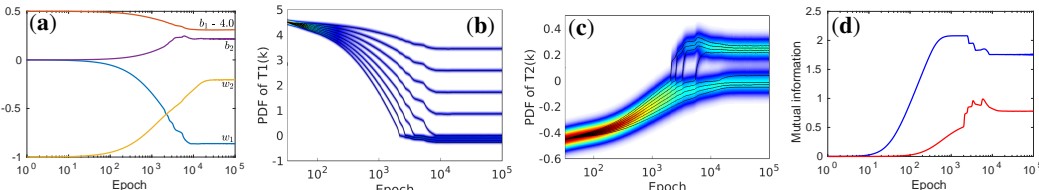

Figure 9: Two-layer leaky ReLU network: (a) network parameters as a function of epoch, (b,c) the corresponding PDFs $p_{T_1(k)}$ and $p_{T_2(k)}$, and (d) the mutual information for both layers.

# 9 EXPERIMENTAL DETAILS

## 9.1 SZT MODEL

In this section we provide additional experimental details and results for the SZT model discussed in Section 5 of the main paper.

To regularize the network weights, we followed (Cisse et al., 2017) and adopted their approach for enforcing an orthonormality constraint. Specifically, we first update the weights $\{W_\ell\}_{\ell \in [L]}$ using the standard gradient descent step, and then perform a secondary update to set

$$W_\ell \leftarrow W_\ell - \alpha \left( W_\ell W_\ell^T - I_{d_\ell} \right) W_\ell,$$

where the regularization parameter $\alpha$ controls the strength of the orthonormality constraint. The value of $\alpha$ was was selected from the set $\{1.0 \times 10^{-5}, 2.0 \times 10^{-5}, 3.0 \times 10^{-5}, 4.0 \times 10^{-5}, 5.0 \times 10^{-5}, 6.0 \times 10^{-5}, 7.0 \times 10^{-5}\}$ and the optimal value was found to be equal to $5.0 \times 10^{-5}$ for both the tanh and ReLU.

In Fig. 10 we present additional experimental results that provide further insight into the clustering and compression phenomena for both tanh and ReLU nonlinearities. Fig. 10(a) shows what happens when the additive noise has a high variance. In this case, although saturation still occurs (see the histograms on top of Fig. 10(a)) and the Gaussians still cluster together (see the scatter plots on the right for the epoch 54 and epoch 8990), compression overall is very mild. The effect of increasing the noise parameter was explained in Section 4 of the main text (see, in particular, Fig. 4(d) therein). Comparing Fig. 10(a) to Fig. 5(a) of the main text, for which $\beta = 0.005$ was used and compression was observed, further highlights the effect of large $\beta$. Recall that smaller $\beta$ values correspond to narrow Gaussians, while larger $\beta$ values correspond to wider Gaussians. When $\beta$ is small, even Gaussians that belong to the same cluster are distinguishable so long as they are not too close. When clusters tighten, the in-class movement brings these Gaussians closer together, effectively merging them, and causing a reduction in mutual information (compression). One the other hand, for large $\beta$, the in-class movement is blurred at the outset (before clusters tighten). Thus, the only effect on mutual information is the separation between the clusters: as these blobs move away from each other, mutual information rises.

Based on the above observation, we can conclude that while the two notions of "clustering Gaussians" and "compression/decrease in mutual information" are strongly related in the low-beta regime, once the noise becomes large, these phenomena decouple, i.e., the network may cluster inputs and neurons may saturate, but this will not be reflected in a decrease of mutual information.

Finally, we present results for ReLU activation without weight normalization (Fig. 10(b)) and with orthonormal weight regularization (Fig. 10(c)). We see that both these networks exhibit almost no compression. For Fig. 10(c), the lack of compression is attributed to regularization of the weight matrices, as explained in Section 5 of the main text. For Fig. 10(b), the reduction in compression can be explained by the fact that although ReLU forces saturation of the neurons at the origin (which promotes clustering), since the positive axes remain unconstrained, the Gaussians can move off towards infinity without bound. This is visible from the histograms in the top row of Fig. 10(b), where, for example, in layer 5 the neurons can take arbitrarily large positive values (note that the bin corresponding to the value 5 accumulates all the values from 5 to infinity). Therefore, the clustering at the origin and the potential drop in mutual information is counterbalanced by the spread of Gaussians along the positive axes and the potential increase of mutual information it causes. Eventually, this leads to the approximately constant profile of the mutual information plot in Fig. 10(b).

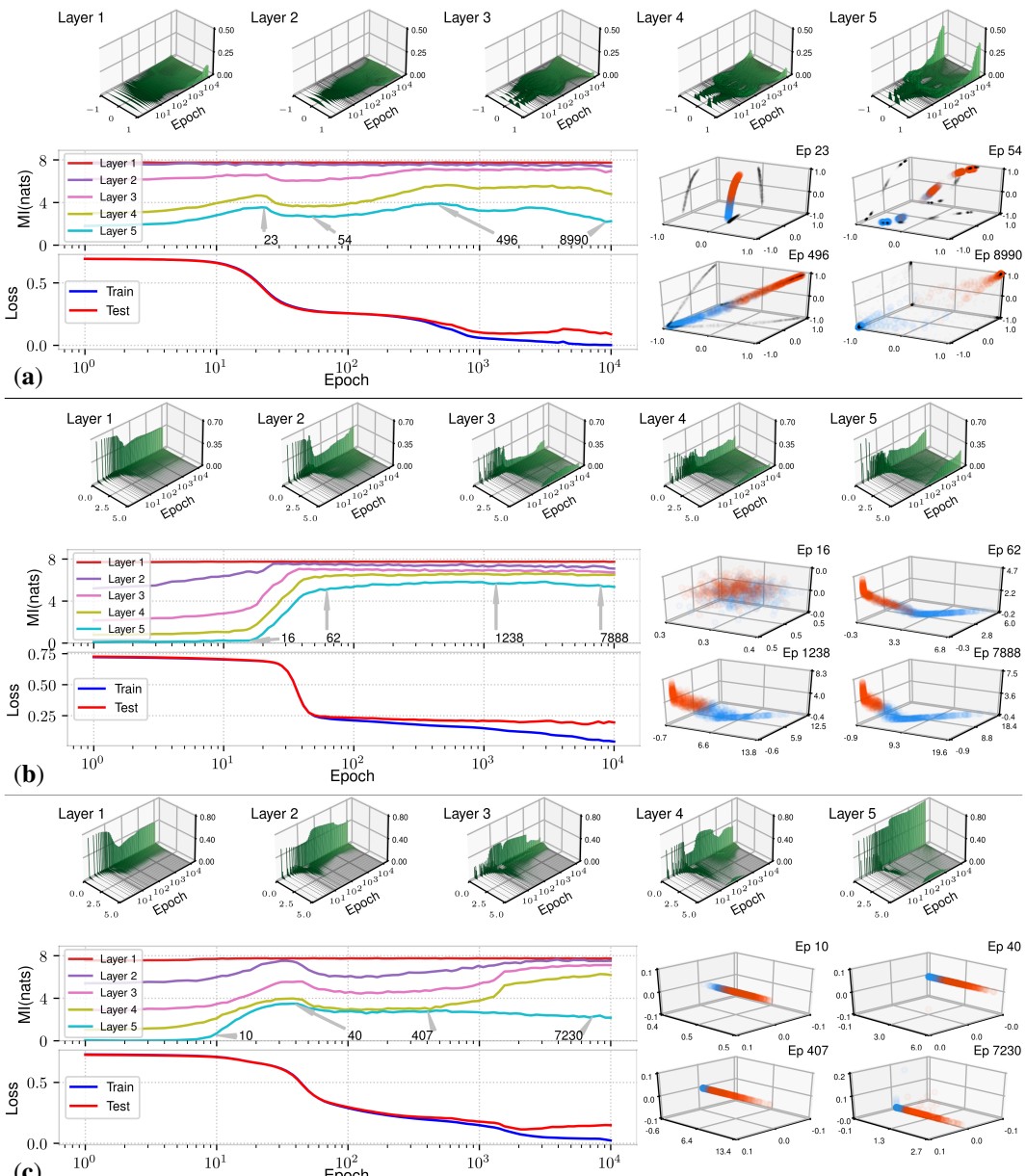

Figure 10: SZT model with (a) tanh nonlinearity and additive noise $\beta = 0.01$ without weight normalization, (b) ReLU nonlinearity and $\beta = 0.01$ without weight normalization, (c) ReLU nonlinearity and $\beta = 0.01$ with weight normalization. Test classification accuracy is 97%, 96%, and 97%, respectively.

The behavior of the weight-normalized ReLU in Fig. 10(c) is similar to Fig. 10(b), although now the growth of the network weights is bounded and the saturation around origin is reduced. For example, for layers 4 and 5 we can see an upward trend in the mutual information, which is then flattened at the end of training. This occurs since more Gaussians are moving away from the origin, although their motion remains bounded (see the histograms on the top and the scatter plots on the right), thus decreasing the clustering density, leading to the rise in the mutual information profile. Once the Gaussians are prevented from moving any further along the positive axes, a slight compression occurs and the mutual information flattens.

## 9.2 Spiral Model

In this section we present results for another synthetic example. We generated data in the form of spiral as in Fig. 11. The network architecture was similar to SZT model, except that the size of each layer was set to 3.

Fig. 12 shows MI estimates $I(X; T_\ell)$ computed using SP estimator and the discrete entropy estimates $H\big(\text{Bin}(T_\ell)\big)$ for weight un-normalized Fig. 12 (a) and normalized models Fig. 12 (b) and using additive noise $\beta = 0.005$. Similar as in the main paper, the results in the figure illustrate a connection between clustering and compression.

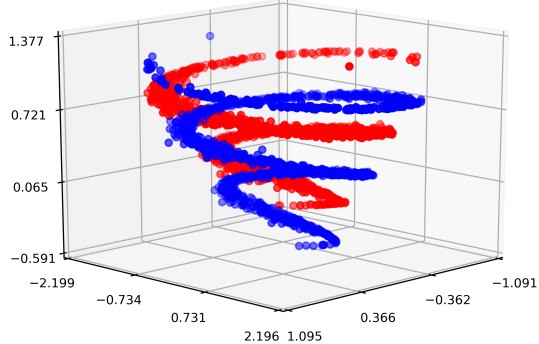

Figure 11: Generated spiral data for binary classification problem.

Finally, in Fig. 13 we also show an estimate of $H\big(\text{Bin}(T_\ell)\big)$ for the case of deterministic DNN trained on spiral data. For the particular choice of the bin size, the result of the estimated entropy reveal a certain level of clustering granularity.

## 9.3 MNIST CNN

In this section, we describe in detail the architecture of the MNIST CNN models used in Sections 2 and 5 in the main paper.

The MNIST CNNs were trained using PyTorch (Paszke et al., 2017) version `0.3.0.post4`. The CNNs use the following fairly standard architecture with two convolutional layers, two fully connected layers, and batch normalization.

1. 2-d convolutional layer with 1 input channel, 16 output channels, 5x5 kernels, and input padding of 2 pixels
2. Batch normalization
3. Tanh() activation function
4. Zero-mean additive Gaussian noise with variance $\beta^2$ or dropout with a dropout probability of 0.2
5. 2x2 max-pooling
6. 2-d convolutional layer with 16 input channels, 32 output channels, 5x5 kernels, and input padding of 2 pixels
7. Batch normalization
8. Tanh() activation function
9. Zero-mean additive Gaussian noise with variance $\beta^2$ or dropout with a dropout probability of 0.2
10. 2x2 max-pooling
11. Fully connected layer with 1586 (32x7x7) inputs and 128 outputs
12. Batch normalization
13. Tanh() activation function

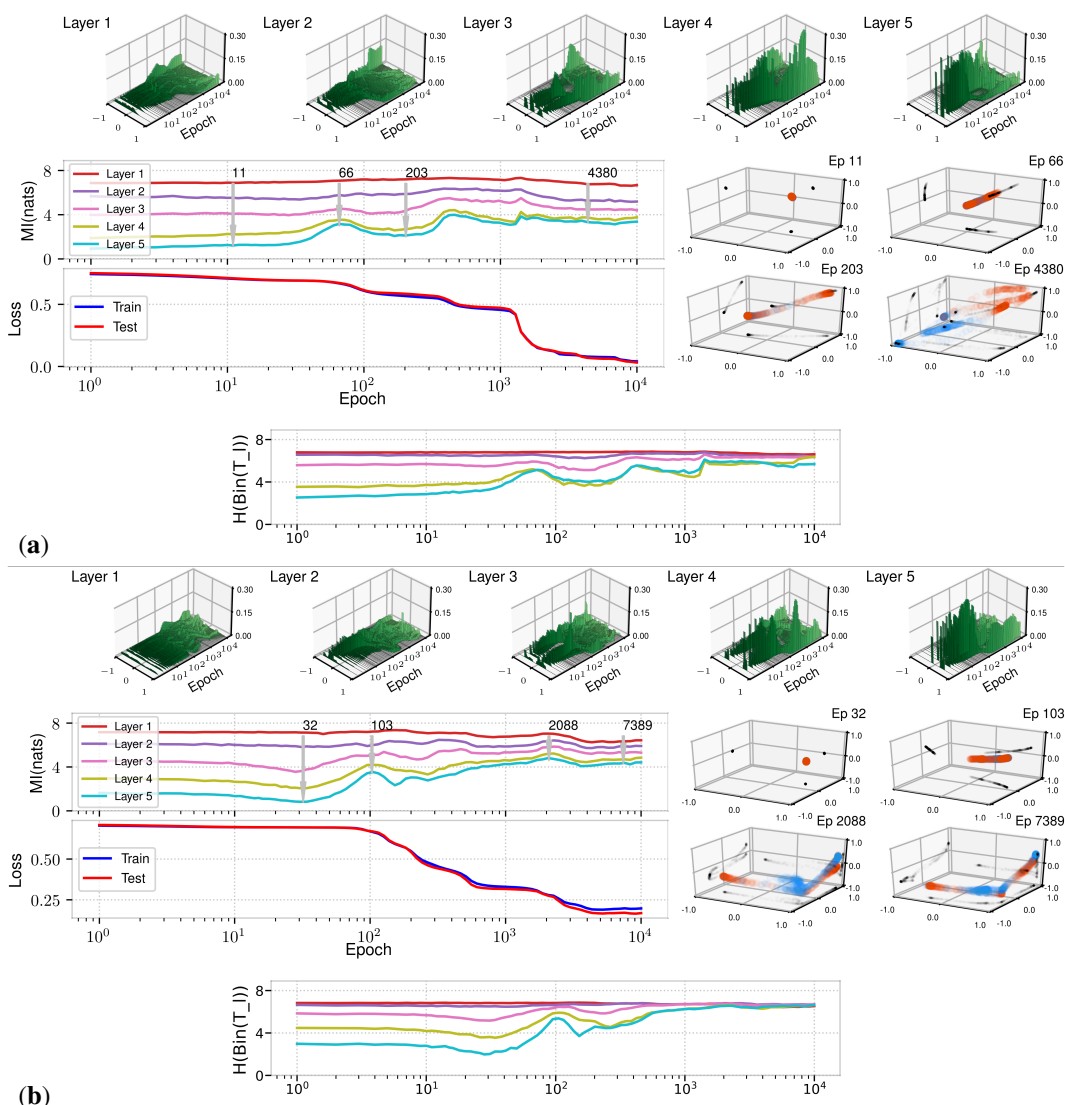

Figure 12: (a) Evolution of $I(X; T_\ell)$ and training/test losses across training epochs for Spiral dataset with $\beta = 0.005$ and tanh nonlinearities. The scatter plots on the right are the values of Layer 5 ($d_5 = 3$) at the arrow-marked epochs on the mutual information plot. The bottom plot shows the entropy estimate $H\big(\text{Bin}(T_\ell)\big)$ across epochs for bin size $B = 10\beta$. (b) Same setup as in (a) but with a regularization that encourages orthonormal weight matrices.

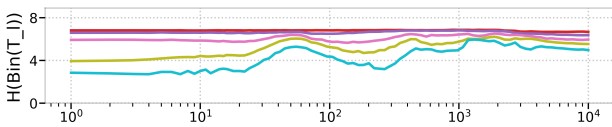

Figure 13: $H\big(\text{Bin}(T_\ell)\big)$ estimate for deterministic net using spiral data. Bin size was set to $B = 0.001$.

14. Zero-mean additive Gaussian noise with variance $\beta^2$ or dropout with a dropout probability of 0.2

15. Fully connected layer with 128 inputs and 10 outputs

All convolutional and fully connected layers have weights and biases, and the weights are initialized using the default initialization, which draws weights from $\mathsf{Unif}[-1/\sqrt{m}, 1/\sqrt{m}]$, with $m$ the fan-in to a neuron in the layer. Training uses cross-entropy loss, and is performed using stochastic gradient descent with no momentum, 128 training epochs, and 32-sample minibatches. The initial learning rate is $5 \times 10^{-3}$, and it is reduced following a geometric schedule such that the learning rate in the final epoch is $5 \times 10^{-4}$. To improve the test set performance of our models, we applied data augmentation to the training set by translating, rotating, and shear-transforming each training example each time it was selected. Translations in the $x$- and $y$-directions were drawn uniformly from $\{-2, -1, 0, 1, 2\}$, rotations were drawn from $\mathsf{Unif}(-10°, 10°)$, and shear transforms were drawn from $\mathsf{Unif}(-10°, 10°)$.

To obtain more reliable performance results, we train eight different models and report the mean number of errors and standard deviation of the number of errors on the MNIST validation set. To ensure that the internal representations of different models are comparable, which is necessary for the use of the cosine similarity measure between internal representations, for each noise condition (deterministic, noisy with $\beta = 0.05$, noisy with $\beta = 0.1$, noisy with $\beta = 0.2$, noisy with $\beta = 0.5$, and dropout with $p = 0.2$), we use a common random seed (different for the eight replications, of course) so the models have the same initial weights and access the training data in the same order (use the same minibatches).

At test time, all models are fully deterministic: the additive noise blocks and dropout layers are replaced by identities. Thus, in the figures and text in the main paper, "Layer 1" is the output of step 5 (2x2 max-pooling), "Layer 2" is the output of step 10 (2x2 max-pooling), "Layer 3" is the output of step 13 (Tanh() activation function), and "Layer 4" is the output of step 15 (fully connected layer with 10 outputs).

## 10    SAMPLE PROPAGATION ESTIMATOR - THEORETIC GUARANTEES

Both conditional and unconditional entropy estimators reduce to the problem of estimating $h(p_S * \varphi)$ using i.i.d. samples $S^n \triangleq (S_i)_{i \in [n]}$ from $S \sim p_S$ while knowing $\varphi$. In this section we state performance guarantees for the SP estimator. These results are excerpted from our work (Anonymized, 2018), where this estimation problem is thoroughly studied. The interested reader is referred to (Anonymized, 2018) for proofs of the subsequently stated results.

### 10.0.1    PRELIMINARY DEFINITIONS

Let $\mathcal{F}_d$ be the set of distributions $P$ with $\mathrm{supp}(P) \subseteq [-1, 1]^d$.[2] The minimax absolute-error risk over $\mathcal{F}_d$ is

$$\mathcal{R}_d^\star(n, \beta) \triangleq \inf_{\hat{h}} \sup_{P \in \mathcal{F}_d} \mathbb{E}_{S^n} \left| h(P * \varphi) - \hat{h}(S^n, \beta) \right|, \tag{9}$$

where $\hat{h}$ is an estimator of $h(P * \varphi)$ based on the empirical data $S^n = (S_1, \dots, S_n)$ of i.i.d. samples from $P$ and the noise parameter $\beta^2$. In (9), by $P * \varphi$ we mean either: (i) $(P * \varphi)(x) = \int_{\mathbb{R}^d} p(u)\varphi(x - u)du = (p * \varphi)(x)$, when $P$ is continuous with density $p$; or (ii) $(P * \varphi)(x) = \sum_{u: \, p(u) > 0} p(u)\varphi(x - u)$, if $P$ is discrete with PMF $p$. This convolved distribution can be defined generally in a way that the two instances above as special cases using measure-theoretic concepts (see (Anonymized, 2018)). Regardless of the nature of $P$, however, we stress that $P * \varphi$ is always a continuous distribution since it corresponds to the random variable $S + Z$, where $Z$ is an isotropic Gaussian. The sample complexity $n_d^\star(\eta, \beta)$ is defined as the smallest number of samples (up to constant factors) for which estimation within an additive gap $\eta$ is possible. Namely,

$$n_d^\star(\eta, \beta) \triangleq \min \left\{ n \big| \mathcal{R}_d^\star(n, \beta) \leq \eta \right\}. \tag{10}$$

We also consider the class of distributions with subgaussian marginals; these will correspond to mutual information estimation over noisy DNNs with ReLU nonlinearities. A subgaussian random variable is defined as follows.

---

[2] Any support included in a compact subset of $\mathbb{R}^d$ would do. We focus on the case of $\mathrm{supp}(P) \subseteq [-1, 1]^d$ due to its correspondence to a noisy DNN with tanh nonlinearities.

**Definition 1 (Subgaussian Random Variable)** *A random variable $X$ is subgaussian if it satisfies either of the following equivalent properties*

1. *Tail condition: $\exists K_1 > 0$, $\mathbb{P}(|X| > t) \leq \exp\left(1 - \frac{t^2}{K_1^2}\right)$, for all $t \geq 0$;*

2. *Moment condition: $\exists K_2 > 0$, $(\mathbb{E}|X|^p)^{\frac{1}{p}} \leq K_2\sqrt{p}$, for all $p \geq 1$;*

3. *Super-exponential moment: $\exists K_3 > 0$, $\mathbb{E}\exp\left(\frac{X^2}{K_3^2}\right) \leq e$,*

*where $K_i$, for $i = 1, 2, 3$, differ by at most an absolute constant. Furthermore, the subgaussian norm $\|X\|_{\psi_2}$ of a subgaussian random variable $X$ is defined as the smallest $K_2$ in property 2, i.e., $\|X\|_{\psi_2} \triangleq \sup_{p \geq 1} p^{-\frac{1}{2}} (\mathbb{E}|X|^p)^{\frac{1}{p}}$.*

Now, let $\mathcal{F}_{d,K}^{(\mathsf{SG})}$ be the class of distributions $P$ of a $d$-dimensional random variable $S = (S(1), \ldots, S(d))$ whose coordinates are subgaussian with $\|S(i)\|_{\psi_2} \leq K$, for all $i \in [d]$. The risk and the sample complexity defined with respect to the nonparametric class $\mathcal{F}_{d,K}^{(\mathsf{SG})}$ are denoted by $\mathcal{R}_{d,K}^{\star}(n, \beta)$ and $n_{d,K}^{\star}(\delta, \beta)$, respectively. Clearly, for any $S \sim P$ with $\text{supp}(P) \subseteq [-1, 1]^d$ we have $\|S(i)\|_{\psi_2} \leq 1$, for all $i \in [d]$, and therefore $\mathcal{F}_d \subseteq \mathcal{F}_{d,1}^{(\mathsf{SG})}$. As a consequence we obtain $\mathcal{R}_d^{\star}(n, \beta) \leq \mathcal{R}_{d,K}^{\star}(n, \beta)$ and $n_d^{\star}(\delta, \beta) \leq n_{d,K}^{\star}(\delta, \beta)$ for all $n \in \mathbb{N}$ and $\delta > 0$, whenever $K \geq 1$. As explained in the following remark, the considered subgaussianity requirement is naturally satisfied by our noisy DNN framework.

**Remark 1 (Generality of Subgaussian Class $\mathcal{F}_{d,K}^{(\mathsf{SG})}$ and Noisy ReLU DNNs)** *The class $\mathcal{F}_{d,K}^{(\mathsf{SG})}$ accounts for distributions induced by noisy DNNs with various nonlinearities. Specifically, it captures the following important cases:*

1. *Distributions with bounded support (corresponding to noisy DNN with bounded activisions).*

2. *Discrete distributions over a finite set, which is a special case of bounded support.*

3. *Distributions of the random variable $S_\ell = f_\ell(T_{\ell-1})$ in a noisy ReLU DNN, so long as the input $X$ to the network is itself subgaussian. To see this recall that linear combinations of independent subgaussian random variables is also subgaussian. Furthermore, for any (scalar) random variable $A$, we have that $|\mathsf{ReLU}(A)| = |\max\{0, A\}| \leq |A|$, almost surely. Now, since each layer in a noisy ReLU DNN is nothing but a coordinate-wise $\mathsf{ReLU}$ applied to a linear transformation of the previous layer plus a Gaussian noise, one may upper bound $(\mathbb{E}|S(i)|^p)^{\frac{1}{p}}$, for a $d_\ell$-dimensional hidden layer $S_\ell$ and $i \in [d_\ell]$, as in Item (2) of Definition 1, provided that the input $X$ is coordinate-wise subgaussian. The constant $K_2$ will depend on the network's weights and biases, the depth of the hidden layer, the subgaussian norm of the input $\|X\|_{\psi_2}$ and the noise variance. This input subgaussianity assumption is, in particular, satisfied by the distribution of $X$ considered herein, i.e., by $X \sim \mathsf{Unif}(\mathcal{X})$.*

## 10.1 Sample Complexity is Exponential in Dimension

We start with two converse claims establishing that the sample complexity is exponential in $d$. The first claim states that there exists a class of distributions $P$, for which the estimation of $h(P * \varphi)$ cannot be done with fewer than exponentially many samples in $d$, when $d$ is sufficiently large.

**Theorem 2 (Asymptotic (in $d$) Exponential Sample-Complexity)** *For any $\beta > 0$ there exist $\gamma(\beta) > 0$ (monotonically decreasing in $\beta$) and a class of distributions in $\mathcal{F}_d$, such that for any $d$ sufficiently large and $\eta > 0$ sufficiently small, the sample complexity of estimating $h(p * \varphi)$ within an additive gap $\eta > 0$ over that class grows as $\Omega\left(\frac{2^{\gamma(\beta)d}}{\left(\eta + \delta_{\beta,d}^{(1)}\right)d}\right)$, where $\lim_{d \to \infty} \delta_{\beta,d}^{(1)} = 0$, for all $\beta > 0$. In particular, $n_d^{\star}(\eta, \beta) = \Omega\left(\frac{2^{\gamma(\beta)d}}{\left(\eta + \delta_{\beta,d}^{(1)}\right)d}\right)$ in this regime.*

The fact that the exponent $\gamma(\beta)$ is monotonically decreasing in $\beta$ suggests that larger values of $\beta$ are favorable for estimation. Theorem 2 shows that an exponential sample complexity is inevitable when $d$ is large. As a complementary result, the next theorem gives a sample complexity lower bound valid in any dimension but only for small enough noise variances. Nonetheless, the result is valid for orders of $\beta$ considered in this work.

**Theorem 3 (Asymptotic (in $\beta$) Exponential Sample-Complexity)** *Fix $d \geq 1$. There exists a class of distributions in $\mathcal{F}_d$ such that for any $\beta, \eta > 0$ sufficiently small, the sample complexity of estimating $h(S + Z)$ within an additive gap $\eta > 0$ over that class grows as $\Omega\left(\frac{2^d}{\left(\eta + \delta_{\beta,d}^{(2)}\right)d}\right)$, where $\lim_{\beta \to 0} \delta_{\beta,d}^{(2)} = 0$, for all $d \geq 1$. In particular, $n_d^\star(\eta, \beta) = \Omega\left(\frac{2^d}{\left(\eta + \delta_{\beta,d}^{(2)}\right)d}\right)$ in this regime.*

**Remark 2** *We state Theorem 3 asymptotically in $\beta$ for the sake of simplicity, but for any $d$ it is possible to follow the constants through the proof to determine a value $c$ such that Theorem 3 holds for all $\beta < c$. For example for $d = 1$, a careful analysis gives that Theorem 3 holds for all $\beta < 0.08$, which is satisfied by most of the experiments run in this paper. This threshold on $\beta$ changes very slowly with increasing $d$ due to the rapid decay of the PDF of the normal distribution.*

## 10.2 Minimax Risk Convergence Rate of the Sample Propagation Estimator

We next focus on analyzing the performance of the SP estimator. For any fixed $S^n = s^n$, denote the empirical PMF associated with $s^n$ by $\hat{p}_{s^n}$. The SP estimator of $h(T) = h(p_S * \varphi)$ is

$$\hat{h}_{\mathsf{SP}}(s^n) \triangleq h(\hat{p}_{s^n} * \varphi). \tag{11}$$

The estimator $\hat{h}_{\mathsf{SP}}(s^n)$ also depends on $\beta$, but we omit this from our notation. The following theorem shows that the expected absolute error of $\hat{h}_{\mathsf{SP}}$ decays like $O\left(\frac{\mathsf{Polylog}(n)}{\sqrt{n}}\right)$ for all dimensions $d$. We provide explicit constants (in terms of $\beta$ and $d$), which present an exponential dependence on the dimension, in accordance to the results of Theorems 2 and 3.

**Theorem 4 (SP Estimator Absolute-Error Risk for Bounded Support)** *Fix $\beta > 0$, $d \geq 1$ and any $\epsilon > 0$. The absolute-error risk of the SP estimator (11) over the class $\mathcal{F}_d$, for all $n$ sufficiently large, is bounded as*

$$\sup_{P \in \mathcal{F}_d} \mathbb{E}_{S^n} \left| h(P * \varphi) - \hat{h}_{\mathsf{SP}}(S^n) \right|$$

$$\leq \frac{1}{2(4\pi\beta^2)^{\frac{d}{4}}} \log\left(\frac{n\left(2 + 2\beta\sqrt{(2+\epsilon)\log n}\right)^d}{(\pi\beta^2)^{\frac{d}{2}}}\right)\left(2 + 2\beta\sqrt{(2+\epsilon)\log n}\right)^{\frac{d}{2}} \frac{1}{\sqrt{n}}$$

$$+ \left(c_{\beta,d}^2 + \frac{2c_{\beta,d}d(1+\beta^2)}{\beta^2} + \frac{8d(d + 2\beta^4 + d\beta^4)}{\beta^4}\right)\frac{2}{n}, \tag{12}$$

*where $c_{\beta,d} \triangleq \frac{d}{2}\log(2\pi\beta^2) + \frac{d}{\beta^2}$. In particular,*

$$\sup_{P \in \mathcal{F}_d} \mathbb{E}_{S^n} \left| h(P * \varphi) - \hat{h}_{\mathsf{SP}}(S^n) \right| = O_{\beta,d}\left(\frac{\mathsf{Polylog}(n)}{\sqrt{n}}\right), \tag{13}$$

*and the right-hand sides (RHSs) of (12) and (13) are, respectively, explicit and implicit upper bounds on the minimax absolute-error risk $\mathcal{R}_d^\star(n, \beta)$.*

**Remark 3 (Comparison to General-Purpose Estimators)** *Note that one could always sample $\varphi$ and add up these noise samples to $S^n$ to obtain a sample set from $P * \varphi$. These samples can be used to get a proxy of $h(P * \varphi)$ via a kNN- or a KDE-based differential entropy estimator. However, $P * \varphi$ violated the boundedness away from zero assumption that most of the convergence rate results in the literature rely on (Levit, 1978; Hall, 1984; Joe, 1989; Hall & Morton, 1993; Tsybakov & der Meulen, 1996; Haje & Golubev, 2009; K et al., 2012; Singh & Póczos, 2016; Kandasamy*

*et al., 2015). The only result we are aware of that analyses a differential entropy estimator (namely, the kNN-based estimator from (A. Kraskov & Grassberger, 2004)) without assuming the density is bounded from below (Jiao et al., 2017) relies on the density being supported inside $[0, 1]^d$, satisfying periodic boundary conditions and having a Hölder smoothness parameter $s \in (0, 2]$. The convolved density $P * \varphi$ satisfies neither of these three conditions. Furthermore, because the SP estimator is constructed to exploit the particular structure of our estimation setup it achieves a fast convergence rate of $\left( \frac{\mathsf{Polylog}(n)}{\sqrt{n}} \right)$. The risk associated with unstructured differential entropy estimators typically converges as the slower $O\left( n^{-\frac{\alpha s}{\beta s + d}} \right)$. This highlights the advantage of ad-hoc estimation as opposed to general-purpose estimation.*

Theorem 1 provides convergence rates when estimating differential entropy (or mutual information) over DNNs with bounded activation functions, such as tanh or sigmoid. To account for networks with unbounded nonlinearities, such as ReLU networks, the following theorem gives a more general result of estimation over the nonparametric class $\mathcal{F}_{d,K}^{(\mathsf{SG})}$ of $d$-dimensional distributions with subgaussian marginals.

**Theorem 5 (SP Estimator Absolute-Error Risk for Subgaussians)** *Fix $\beta > 0$ and $d \geq 1$. The absolute-error risk of the SP estimator (11) over the class $\mathcal{F}_{d,K}^{(\mathsf{SG})}$, for all $n$ sufficiently large, is bounded as*

$$
\sup_{P \in \mathcal{F}_{d,K}^{(\mathsf{SG})}} \mathbb{E}_{S^n} \left| h(P * \varphi) - \hat{h}_{\mathsf{SP}}(S^n) \right| \leq \log \left( \sqrt{n} \left( \frac{8(K + \beta)^2}{(e-1)\pi\beta^2} \right)^{\frac{d}{4}} \right) \left( \frac{2(K+\beta)^2}{(e-1)\pi\beta^2} \log n \right)^{\frac{d}{4}} \frac{1}{\sqrt{n}}
$$

$$
+ \left( \left( c_{\beta,d}' + 2dK^2 \right)^2 + \frac{2\left( c_{\beta,d}' + 2dK^2 \right)\left( \beta^2 d + 2dK^2 \right)}{\beta^2} + \frac{8\left( 16d^2 K^4 + \beta^4 d(2 + d) \right)}{\beta^4} \right) \frac{ed}{n},
$$

$$(14)$$

*where $c_{\beta,d}' \triangleq \frac{d}{2} \log(2\pi\beta^2)$. In particular,*

$$
\sup_{P \in \mathcal{F}_{d,K}^{(\mathsf{SG})}} \mathbb{E}_{S^n} \left| h(P * \varphi) - \hat{h}_{\mathsf{SP}}(S^n) \right| = O_{\beta,d} \left( \frac{\mathsf{Polylog}(n)}{\sqrt{n}} \right), \tag{15}
$$

*and the RHSs of (14) and (15) are, respectively, explicit and implicit upper bounds on the minimax absolute-error risk $\mathcal{R}_{d,K}^\star(n, \beta)$.*

As mentioned in Remark 1, the class $\mathcal{F}_{d,K}^{(\mathsf{SG})}$ is rather general, and, in particular, includes $\mathcal{F}_d$ whenever $K \geq 1$. This means that Theorem 5 also provides an upper bound on the minimax risk under the setup of Theorem 1. Nonetheless, we chose to separately state Theorem 1 since the derivation under the bounded support assumption enables extracting slightly better constants (which is important for our applications - see Section 5). We do highlight, however, that the expressions from (12) and (14) with $K = 1$ not only have the same convergence rates, but their constants are also very close.

**Remark 4 (Near Minimax Rate-Optimality)** *A convergence rate faster than $\frac{1}{\sqrt{n}}$ cannot be attained for parameter estimation under the absolute-error loss. This follows from, e.g., Proposition 1 of (Chen, 1997), which establishes this convergence rate as a lower bound for the parametric estimation problem given $n$ i.i.d. samples. Consequently, the convergence rate of $O_{\sigma,d}\left( \frac{\mathsf{Polylog}(n)}{\sqrt{n}} \right)$ established in Theorems 1 and 5 for the SP estimator is near minimax rate-optimal (i.e., up to logarithmic factors).*

**Remark 5 (Mutual Information Estimation)** *Denoting the upper bound on the estimation error from Theorem 1 or 5 by $\Delta_n(\beta, d)$, we see that the error of the mutual information estimator from (2) is bounded as by $2\Delta_n(\beta, d)$, which vanishes as $n \to \infty$.*

### 10.2.1 SAMPLE PROPAGATION ESTIMATOR BIAS

The results of the previous subsection are of minimax flavor. That is, they state worst-case convergence rates of the SP estimation over a certain nonparametric class of distributions. In practice, the true distribution may very well not be one that attains these worst-case rates, and convergence may be faster. However, while variance of $\hat{h}_{\mathsf{SP}}(S^n)$ can be empirically evaluated using bootstrapping, there is no empirical test for the bias. Even if multiple estimations of $h(P * \varphi)$ via $\hat{h}_{\mathsf{SP}}(S^n)$ consistently produce similar values, this does not necessarily suggest that these values are close to the true $h(P * \varphi)$. To have a guideline to the least number of samples needed to avoid biased estimation, we present the following lower bound on $\sup_{P \in \mathcal{F}_d} \mathbb{E}_{S^n} \left| h(P * \varphi) - \hat{h}_{\mathsf{SP}}(S^n) \right|$.

**Theorem 6 (SP Estimator Bias Lower Bound)** *Fix $d \geq 1$ and $\beta > 0$, and let $\epsilon \in \left(1 - \left(1 - 2Q\left(\frac{1}{2\beta}\right)\right)^d, 1\right]$, where $Q$ is the Q-function.[3] Set $k_\star \triangleq \left\lfloor \frac{1}{\beta Q^{-1}\left(\frac{1}{2}\left(1 - (1-\epsilon)^{\frac{1}{d}}\right)\right)} \right\rfloor$, where $Q^{-1}$ is the inverse of the Q-function. By the choice of $\epsilon$, clearly $k_\star \geq 2$, and the bias of the SP estimator over the class $\mathcal{F}_d$ is bounded as*

$$\left| \sup_{P \in \mathcal{F}_d} h(P * \varphi) - \mathbb{E}_{S^n} \hat{h}_{\mathsf{SP}}(S^n) \right| \geq \log\left( \frac{k_\star^{d(1-\epsilon)}}{n} \right) - H_b(\epsilon). \quad (16)$$

*Consequently, the bias cannot be less than a given $\delta > 0$ so long as $n \leq k_\star^{d(1-\epsilon)} \cdot e^{-(\delta + H_b(\epsilon))}$.*

Since $H_b(\epsilon)$ shrinks with $\epsilon$, for sufficiently small $\epsilon$ values the lower bound from (16) essentially shows that the SP estimator will not have negligible bias unless $n > k_\star^{d(1-\epsilon)}$ is satisfied. The condition $\epsilon > 1 - \left(1 - 2Q\left(\frac{1}{2\beta}\right)\right)^d$ is non-restrictive in any relevant regime of $\beta$ and $d$. For instance, for typical $\beta$ values we work with - around $0.1$ - this lower bound is at most $0.0057$ for all dimensions up to at least $d = 10^4$. Setting, e.g., $\epsilon = 0.01$ (for which $H_b(0.01) \approx 0.056$), the corresponding $k_\star$ equals 3 for $d \leq 11$ and 2 for $12 \leq d \leq 10^4$. Thus, with these parameters, the number of estimation samples $n$ should be at least $2^{0.99d}$, for any conceivably relevant dimension, in order to have negligible bias.

### 10.3 COMPUTING THE SAMPLE PROPAGATION ESTIMATOR

Evaluating the mutual information estimator from (2) requires computing the differential entropy of a Gaussian mixture. Although it cannot be computed in closed form, this section presents a method for approximate computation via MC integration (Robert, 2004). To simplify the presentation, we present the method for an arbitrary Gaussian mixture without referring to the notation of the estimation setup.

Let $g(t) \triangleq \frac{1}{n} \sum_{i \in [n]} \varphi(t - \mu_i)$ be a $d$-dimensional, $n$-mode Gaussian mixture, with $\{\mu_i\}_{i \in [n]} \subset \mathbb{R}^d$ and $\varphi$ as the PDF of $\mathcal{N}(0, \beta^2 \mathrm{I}_d)$. Let $C \sim \mathsf{Unif}\{\mu_i\}_{i \in [n]}$ be independent of $Z \sim \varphi$ and note that $V \triangleq C + Z \sim g$.

We use Monte Carlo (MC) integration (Robert, 2004) to compute the $h(g)$. First note that

$$h(g) = -\mathbb{E} \log g(V) = -\frac{1}{n} \sum_{i \in [n]} \mathbb{E}\left[ \log g(\mu_i + Z) \Big| C = \mu_i \right] = -\frac{1}{n} \sum_{i \in [n]} \mathbb{E} \log g(\mu_i + Z), \quad (17)$$

where the last step follows by the independence of $Z$ and $C$. Let $\left\{ Z_j^{(i)} \right\}_{\substack{i \in [n] \\ j \in [n_{\mathsf{MC}}]}}$ be $n \times n_{\mathsf{MC}}$ i.i.d. samples from $\varphi$. For each $i \in [n]$, we estimate the $i$-th summand on the RHS of (17) by

$$\hat{I}_{\mathsf{MC}}^{(i)} \triangleq \frac{1}{n_{\mathsf{MC}}} \sum_{j \in [n_{\mathsf{MC}}]} \log g\left( \mu_i + Z_j^{(i)} \right), \quad (18a)$$

---

[3] The Q-function is defined as $Q(x) \triangleq \frac{1}{\sqrt{2\pi}} \int_x^\infty e^{-\frac{t^2}{2}} dt$.

which produces

$$\hat{h}_{\mathsf{MC}} \triangleq \frac{1}{n} \sum_{i \in [n]} \hat{I}_{\mathsf{MC}}^{(i)} \tag{18b}$$

as our estimate of $h(g)$. Define the mean squared error (MSE) of $\hat{h}_{\mathsf{MC}}$ as

$$\mathsf{MSE}\left(\hat{h}_{\mathsf{MC}}\right) \triangleq \mathbb{E}\left[\left(\hat{h}_{\mathsf{MC}} - h(g)\right)^2\right]. \tag{19}$$

We have the following bounds on the MSE for tanh and ReLU networks.

**Theorem 7 (MSE Bounds for MC Estimator)**

(i) *Assume $C \in [-1, 1]^d$ almost surely (i.e., tanh network), then*

$$\mathsf{MSE}\left(\hat{h}_{\mathsf{MC}}\right) \leq \frac{1}{n \cdot n_{\mathsf{MC}}} \frac{2d(2 + \beta^2)}{\beta^2}. \tag{20}$$

(ii) *Assume $M_C \triangleq \mathbb{E}\|C\|_2^2 < \infty$ (e.g., ReLU network with bounded 2nd moments), then*

$$\mathsf{MSE}\left(\hat{h}_{\mathsf{MC}}\right) \leq \frac{1}{n \cdot n_{\mathsf{MC}}} \frac{9d\beta^2 + 8(2 + \beta\sqrt{d})M_C + 3(11\beta\sqrt{d} + 1)\sqrt{M_C}}{\beta^2}. \tag{21}$$

The bounds on the MSE scale only linearly with the dimension $d$, making $\sigma^2$ in the denominator often the dominating factor experimentally.

**Remark 6 (Comparison to Generic Entropy Estimation)** *We briefly present empirical results illustrating the convergence of the SP estimator and comparing it to two current state-of-the-art methods: the KDE-based estimator of (Kandasamy et al., 2015) and the kNN-based estimator often known as the Kozachenko–Leonenko (KL) nearest neighbor estimator (Kozachenko & Leonenko, 1987; Jiao et al., 2017). In this example, the distribution $P$ of $S$ is set to be a mixture of Gaussians truncated to have support in $[-1, 1]^d$. Before truncation, the mixture consists of $2^d$ Gaussian components with means at the $2^d$ corners of $[-1, 1]^d$. The entropy of $P * \phi$, i.e., $h(S + Z)$, where $Z \sim \mathcal{N}(0, \sigma^2 \mathrm{I}_d)$, is estimated and various values of $\sigma$ are examined.*

*Fig. 14 shows the estimation error results as a function of the number of samples $n$ for dimensions $d = 5$ and $d = 10$. The kernel width for the KDE estimate was chosen via cross-validation, varying with both $d$ and $n$; the kNN estimator and $\hat{h}_{\mathsf{SP}}(S^n)$ require no tuning parameters. We found that the KDE estimate is highly sensitive to the choice of kernel width, the curves shown correspond to optimized values and are highly unstable to any change in kernel width. Note that both the kNN and the KDE estimators converge slowly, at a rate that degrades with increased $d$. This rate is worse than that of $\hat{h}_{\mathsf{SP}}$, which also lower bounds the true entropy (as according to our theory - see (Anonymized, 2018, Equation (60))).*

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

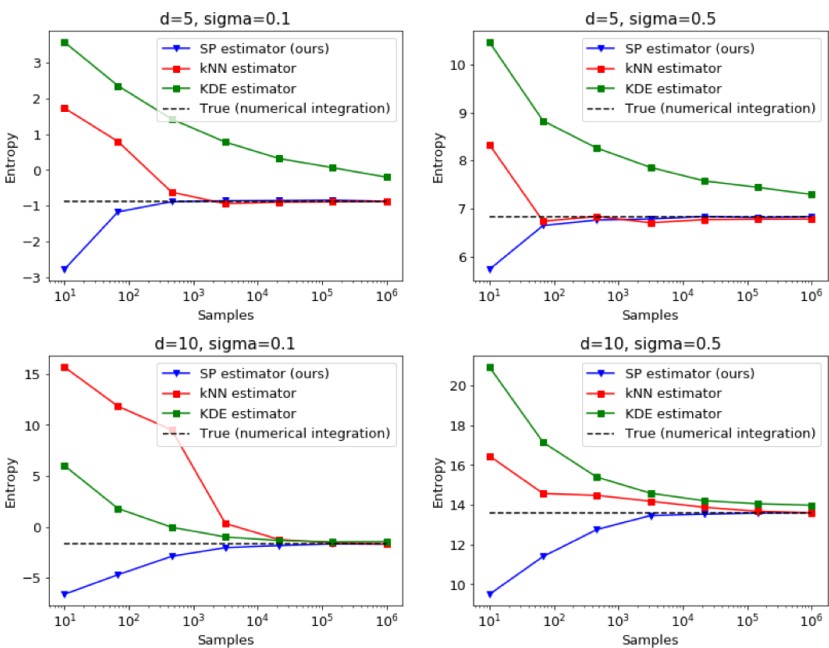

Figure 14: Estimation results for the SP estimator compared to state-of-the-art kNN-based and KDE-based differential entropy estimators. The differential entropy of $S + Z$ is estimated, where $S$ is a truncated $d$-dimensional mixture of $2^d$ Gaussians and $Z \sim \mathcal{N}(0, \sigma^2 \mathrm{I}_d)$. Results are shown as a function of $n$, for $d = 5, 10$ and $\sigma = 0.1, 0.5$. The SP estimator presents faster convergence rates, improved stability and better scalability with dimension compared to the two competing methods.

P. Hall. Limit theorems for sums of general functions of m-spacings. *Mathematical Proceedings of the Cambridge Philosophical Society*, 96(3):517–532, Nov. 1984.

P. Hall and S. C. Morton. On the estimation of entropy. *Annals of the Institute of Statistical Mathematics*, 45(1):69–88, Mar. 1993.

J. Jiao, W. Gao, and Y. Han. The nearest neighbor information estimator is adaptively near minimax rate-optimal. arXiv:1711.08824 [stat.ML], 2017.

H. Joe. Estimation of entropy and other functionals of a multivariate density. *Annals of the Institute of Statistical Mathematics*, 41(4):683–697, Dec. 1989.

Sricharan K, R. Raich, and A. O. Hero. Estimation of nonlinear functionals of densities with confidence. *IEEE Trans. Inf. Theory*, 58(7):4135–4159, Jul. 2012.

K. Kandasamy, A. Krishnamurthy, B. Poczos, L. Wasserman, and J. M. Robins. Nonparametric von Mises estimators for entropies, divergences and mutual informations. In *Advances in Neural Information Processing Systems (NIPS)*, pp. 397–405, 2015.

L. F. Kozachenko and N. N. Leonenko. Sample estimate of the entropy of a random vector. *Problemy Peredachi Informatsii*, 23(2):9–16, 1987.

B. Y. Levit. Asymptotically efficient estimation of nonlinear functionals. *Problemy Peredachi Informatsii*, 14(3):65–72, 1978.

A. Paszke, S. Gross, S. Chintala, G. Chanan, E. Yang, Z. DeVito, Z. Lin, A. Desmaison, L. Antiga, and A. Lerer. Automatic differentiation in PyTorch. In *NIPS Autodiff Workshop*, 2017.

Christian P Robert. *Monte Carlo Methods*. Wiley Online Library, 2004.

A. M. Saxe, Y. Bansal, J. Dapello, M. Advani, A. Kolchinsky, B. D. Tracey, and D. D. Cox. On the information bottleneck theory of deep learning. In *Proceedings of the International Conference on Learning Representations (ICLR)*, 2018.

S. Singh and B. Póczos. Finite-sample analysis of fixed-k nearest neighbor density functional estimators. In *Advances in Neural Information Processing Systems*, pp. 1217–1225, 2016.

A. B. Tsybakov and E. C. Van der Meulen. Root-$n$ consistent estimators of entropy for densities with unbounded support. *Scandinavian Journal of Statistics*, pp. 75–83, Mar. 1996.

