# OpenReview forum: "Estimating Information Flow in DNNs"
_ICLR.cc/2019/Conference_

### Official Review · AnonReviewer3 · 2018-10-16
**ICLR 2019 Conference Paper636 AnonReviewer3**

**Rating:** 4
**Confidence:** 5

**Review:**

This paper studied the information bottleneck principle for deep learning. In the paper by (Schwatz-Ziv & Tishby 17'), it is empirically shown that the mutual information I(X;T) between input X and internal layers T decreases, which is called a compression phase. In this paper, the author found that the compression phase is not always happening and the shape of the curve of I(X;T) highly depends on the "bining size" which is used for estimating mutual information by (Schwatz-Ziv & Tishby 17'). Then the authors proposed to use a noisy DNN to make sure the map X->T is stochastic, then proposed a guaranteed mutual information estimator. Then some empirical results are shown.

I think the problem in (Schwatz-Ziv & Tishby 17') do exist and their result is highly questionable. However, I have some major question about this paper.

1. In this paper a noisy DNN was proposed. However, how do you choose the noise level \beta? If I understand correctly, the noise level plays a similar role of the bining size in (Schwatz-Ziv & Tishby 17'). Noise level goes to zero is similar to bining size goes to zero. I wish to see a figure about how different \beta affects the curve of I(X;T) (similar to Figure 1 but let \bet change).

    In Figure 4(d) there is a plot showing different \beta will affect the mutual information, but the x-axis is "weight". I wonder that how the curve of mutual information change w.r.t \beta, if the x-axis is training epochs. Do your statement stable about \beta?

2. I think Section 3 and Theorem 1 are interesting and insightful. But I notice that in Section 10 you mentioned that this will be a separate paper. Is it OK to put them together in this paper?

3. The paper by (Schwatz-Ziv & Tishby 17') has not pass a peer-review process and it is still a preprint. This paper is nothing but only saying some deficiencies of (Schwatz-Ziv & Tishby 17') (except Section 3 and Theorem 1 which I think should be an independent paper). I think such a paper should not be published as a conference paper before (Schwatz-Ziv & Tishby 17') pass a peer-review process.

So totally I think this paper should not be accepted by ICLR at this point. I think Section 3 and Theorem 1 should become an independent paper, and the DNN approach can be an application of the mutual information estimator.

---

> ### Author Response · Authors · 2018-11-09
> **Response to Reviewer 3 (part 1)**
>
> "However, how do you choose the noise level \beta?"
>
> Ideally, \beta should be treated as a hyperparameter that is selected to optimize the performance of the classifier on held-out data, much in the way that hyperparameters such as dropout rate are tuned to optimize held-out performance. In practice, we sometimes had to back off from the \beta value that optimizes performance to a higher value to ensure accurate estimation of mutual information (the smaller \beta is the harder it becomes to estimate I(X:T_\ell)), depending on factors such as the dimensionality of the layer being analyzed and the number of data samples available for a task. The bounds described in the Supplement were used to provide guidance on values of \beta that the estimator can handle using a given number of samples.
>
>
> "If I understand correctly, the noise level plays a similar role of the bining size in (Schwatz-Ziv & Tishby 17'). Noise level goes to zero is similar to bining size goes to zero."
>
> The noisy setting and binning-based estimation in deterministic DNNs are fundamentally different. Binning is a *method for* differential entropy (and thus mutual information) *estimation* that has no theoretic convergence guarantees that we are aware of when the bin size is fixed. In deterministic DNNs, I(X;T_\ell)=H(X) is constant and any plot that shows otherwise shows a faulty estimate. While tweaking the bin sizes *changes the estimate* and the plot (see Fig. 1), the true mutual information remains constant (=H(X)). In contrast, the noise parameter \beta affects the true mutual information in our noisy DNN (because the noise is part of the DNN's operation), as shown in Fig. 4(d). We emphasize that \beta primarily affects the degree to which I(X;T_\ell) is affected by the underlying clustering trend (observed compression is more pronounced for smaller \beta, and disappears for very large \beta). The main point here, however, is that trends shown in our mutual information plots track the true behavior of I(X;T_\ell) in the noisy DNN (as suggested by our theoretical estimation risk bound), while binning-based estimation of I(X;T_\ell) in a deterministic DNN produces plots in which the curves vary only due to estimation errors. We note that if the network from (Shwartz-Ziv & Tishby, 2017) had quantizers applied on the outputs of the neurons (which it does not), then the choice of the quantization gap (i.e., the bin size) would have been analogous to the \beta parameter in our work.
>
>
> "I wish to see a figure about how different \beta affects the curve of I(X;T) (similar to Figure 1 but let \bet change)?"
>
> At the end of Section 4 we explain how different \beta values affect the observed relation between clustering and compression. Basically, for larger \beta values the Gaussians at the output of the noisy neuron are indistinguishable to begin with, and consequently, clustering the internal representations has less effect on the mutual information. Thus, I(X:T_\ell) is better for tracking clustering for smaller values of \beta. The revised Fig. 4(d) visualizes the effect of \beta on the mutual information in the minimal example, and Figs. 5(a) and 10(a) show the same tanh experiment for the noisy version of the DNN from (Shwartz-Ziv & Tishby, 2017) with different \beta values (0.005 and 0.01, respectively). As claimed, smaller \beta makes compression less pronounced.
>
>
> "In Figure 4(d) there is a plot showing how different \beta will affect the mutual information, but the x-axis is "weight". I wonder that how the curve of mutual information change w.r.t \beta, if the x-axis is training epochs. Do your statement stable about \beta?"
>
> Thanks for the suggestion. We have updated Fig. 4(d) to have the x-axis be training epochs as requested. The results show the desired stability with respect to \beta. We note that we originally presented the mutual information curve in Fig. 4(d) vs a growing weight parameter since, in this minimal example, the weight monotonically grows throughout training. Indeed, the original Fig. 4(d) and the revised one in the current version of the text look very much alike, up to some horizontal stretching of the curves.

---

> ### Author Response · Authors · 2018-11-09
> **Response to Reviewer 3 (part 2)**
>
> "I think Section 3 and Theorem 1 are interesting and insightful. But I notice that in Section 10 you mentioned that this will be a separate paper. Is it OK to put them together in this paper?"
>
> We think the reviewer is asking why we did not include the full proof of Theorem 1 and related theory in the ICLR submission. The answer is that including all of the theory and the empirical work would require nearly 30 pages, while the ICLR page limit is 10 pages. We thus had to split the theoretical work and empirical work into two papers. In the interest of transparency, we included the key parts of the theoretical work in the supplement and explained that there is a parallel paper (in review) on the theory. Note, however, that Section 3 and Theorem 1 will remain in the final version of the ICLR paper (if it is accepted). Our original plan was to omit Section 10 of the current supplement, replacing it with a non-anonymized citation of the companion paper. However, if the reviewer thinks that it would be better to keep Section 10 in the final version, we are happy to comply with that suggestion.
>
>
> "The paper by (Schwatz-Ziv & Tishby 17') has not pass a peer-review process and it is still a preprint. This paper is nothing but only saying some deficiencies of (Schwatz-Ziv & Tishby 17') (except Section 3 and Theorem 1 which I think should be an independent paper). I think such a paper should not be published as a conference paper before (Schwatz-Ziv & Tishby 17') pass a peer-review process."
>
>
> According to Google Scholar, (Shwartz-Ziv & Tishby, 2017) has 178 citations. Naftali Tishby's lecture from the "Deep Learning: Theory, Algorithms, and Applications" workshop held in June 2017 in Berlin has over 70,000 views on YouTube. This work, even if it has not appeared in a peer-reviewed venue, has received plenty of attention in the deep learning community. It is therefore an appropriate subject for other scholarly work.
>
>
> Indeed, many of the issues we identify with the (Shwartz-Ziv & Tishby, 2017) analysis also appear in other *published* works that we cite, e.g. (Saxe et al., 2018, published at ICLR 2018). We focus our discussion on (Shwartz-Ziv & Tishby, 2017) because it is the most well-known and indeed was the first work in this area. However, we could have just as easily chosen one of the other peer reviewed works we cite as the focus of our discussion.

---

### Official Review · AnonReviewer1 · 2018-10-28
**An interesting paper but the observations from the experiments could be stated more clear.**

**Rating:** 7
**Confidence:** 4

**Review:**

Response to author comments:

I would like to thank the authors for answering my questions and addressing the issues in their paper. I believe the edits and newly added comments improve the paper.

I found the response regarding the use of your convergence bound very clear. It is a very reasonable use of the bound and now I see how you take advantage of it in your experimental work. However, I believe the description in the paper, in particular, the last two sentences of Remark 1, could still be improved and better explain how a reasonable and computationally feasible n was chosen.

To clarify one of my questions, you correctly assumed that I meant to write the true label, and not the output of the network.


***********

The paper revises the techniques used in Tishby’s and Saxe et al. work to measure mutual information between the data and a hidden layer of a neural network. The authors point out that these previous papers’ measures of mutual information are not meaningful due to lack of clear theoretical assumptions on the randomness that arises in DNNs.

The authors propose to study a perturbed version of a neural network to turn it into a noisy channel making the mutual information estimation meaningful. The perturbed network has isotropic Gaussian noise added to each layer nodes. The authors then propose a method to estimate the mutual information of interest. They suggest that the mutual information describes how distinguishable the hidden representation values are after a Gaussian perturbation (which is equivalent to estimating the means of a mixture of Gaussians). Data clustering per class is identified as the source of compression.

In addition to proposing a way to estimate a mutual information of a stochastic network, the authors analyze the compression that occurs in stochastic neural networks.

It seems that the contribution is empirical, rather than theoretical, as the theoretical result cited is going to appear in a different article. After reading that the authors “develop sample propagation (SP) estimator”, I expected to see a novel approach/algorithm. However, unless I missed something, the proposed method for estimating MI for this Gaussian channel is just doing MC estimation (and no guarantees are established in this paper). The convergence bounds for the SP estimator are presented(Theorem 1), however, the result is cited from another article of the authors, so it is not a contribution of this submission.

Since the authors have this convergence  bound stated in Theorem 1, it would be great to see it being used - how many samples are needed/being used in the experiments? What should the error bars be around mutual information estimates in the experiments? If the bound is too loose for a reasonable number of samples, then what’s the use of it?

The authors perform two types of experiments on MNIST. The first experiment demonstrates that no compression is observed per layer and the mutual information only increases during training (as measured by the binning approach, which is supposed to track the mutual information of the stochastic version of the network). The second experiments demonstrates that deeper layers perform more clustering.

Regarding the first experiment, could the authors clarify how per unit and per entire layer compression estimation differs?

Also, in my opinion, more clustered representations seem to indicate that the mutual information with the output increases. Could the authors comment on how the noise levels in this particular version of a stochastic network affects the mutual information with the output and the clustering? Do more clustered representations lead to increased mutual information of the layer with the output?

I found it fairly difficult to summarize the experimental contribution after the first read. I think the presentation and summary after each experiment could be improved and made more reader friendly. For example, the authors could include a short section before the experiments stating their hypothesis and pointing to the experiment/figure number supporting their hypothesis.

---

> ### Author Response · Authors · 2018-11-09
> **Response to Reviewer 1 (part 1)**
>
> "Since the authors have this convergence bound stated in Theorem 1, it would be great to see it being used - how many samples are needed/being used in the experiments? What should the error bars be around mutual information estimates in the experiments? If the bound is too loose for a reasonable number of samples, then what’s the use of it?"
>
> Thank you for this very relevant comment. We first note that Theorem 1 provides a worst-case result: it bounds the absolute-error risk of the differential entropy estimation given the worst possible probability distribution. What this means in practice is that the bound is quite pessimistic because the distributions induced by the DNN generally do not follow the pathological structures that attains the worst case bound.
>
> We emphasize that the theoretical bound is still worthwhile. From a theoretical perspective it gives justification for applying our estimator in the noisy DNN setup and in other problems, and a guideline for determining the approximate highest dimensionality that we can handle. From a practical perspective it gives a worst-case starting point for the number of samples, n, which can be reduced if the estimator empirically performs better than worst-case.
>
> This is, in fact, how the bound was used for producing our simulation results. Generating the curves in our plots required running the sample-propagation differential entropy estimator multiple times. First, estimating the mutual information term of interest I(X;T_\ell) for a given set of DNN parameters involves computing m+1 differential entropy estimates, where m is the size of the empirical dataset. Then, we had to estimate I(X;T_\ell) not just once but for each epoch of training. To overcome this computational burden while adhering to the theoretical result, we tested the value of n given by Theorem 1 on a few points of the curve and reduced it until the overall computation cost of producing the full curve became reasonable. To ensure estimation accuracy was not compromised we empirically tested that the estimate remained stable.
>
> As a concrete example, to achieve an error bound of 5% of Fig. 5 plot's vertical scale (which amounts to an 0.4 absolute error bound), the number of samples required by Theorem 1 is n=4*10^9. This number is too large for our computational budget. Performing the above procedure for reducing n, we find good accuracy is achieved for n = 4*10^6 samples (Theorem 1 has the pessimistic error bound of 3.74 for this value). Adding more samples beyond this value does not change the results.

---

> ### Author Response · Authors · 2018-11-09
> **Response to Reviewer 1 (part 2)**
>
> "[U]nless I missed something, the proposed method for estimating MI for this Gaussian channel is just doing MC estimation..."
>
> The noisy neural network channel is not a Gaussian channel because it involves a composition of multiple layers of nonlinearities and Gaussian noises that the input signal has to traverse until it reaches layer \ell. Writing T_\ell=S_\ell+Z_\ell, with S_\ell=f_\ell(T_{\ell-1}), this concatenation of nonlinear operations and Gaussians renders the distributions of S_\ell (marginal or conditioned on X) extremely complicated. Not only can these distributions not be written out in an analytic form, they are even *extremely* hard to numerically evaluate at any given point. Our best mode of operation was therefore to treat P_{S_\ell} and P_{S_\ell|X} as unknown. However, the generative model of the DNN does permit us to efficiently sample from P_{S_\ell} and P_{S_\ell|X}, which brings us to the considered functional estimation problem: estimating the differential entropy h(P_{S_\ell}\ast\gamma) based on i.i.d. samples from the *unknown* distribution P_{S_\ell} and knowledge of the noise distribution \gamma (this can be equivalently viewed as the estimation of the functional T_\gamma(P_{S_\ell})\triangleq h(P_{S_\ell}\ast\gamma) of the unknown P_{S_\ell} based on i.i.d. samples from it).
>
> We note that it is not possible to simply apply Monte Carlo integration to estimate the differential entropy h(Q) of an unknown distribution Q using only i.i.d. samples from Q: the MC integrator would also need to know Q itself. The crux of differential entropy estimation is to find a function of the samples alone that approximates h(Q). In our case of Q=P_{S_\ell}\ast\gamma, the SP estimator uses the samples from P_{S_\ell} and the known noise distribution to form a provably consistent estimate of the entropy that is expressed in terms of a d-dimensional integral. Because this integral cannot be evaluated in closed form, we use MC integration *merely to evaluate the integral*.
>
> We clarify the full estimation process and the role of each component next:
>
> (i) Expand I(X;T_\ell)=h(T_\ell)-\frac{1}{m}\sum_{i=1}^m h(T_\ell|X=x_i).
>
> (ii) Since T_\ell=S_\ell+Z_\ell and S_\ell and Z_\ell are independent, the distribution of T_\ell is P_{S_\ell} \ast \gamma. We know \gamma since the noise is injected by design, and we can sample from P_{S_\ell} via the DNN's forward pass. Estimating I(X;T_\ell) reduces to a new functional estimation problem: estimate h(A+B) given i.i.d. samples from A and knowing the distribution of B ~ N(0,\beta^2 I_d).
>
> (iii) SP Estimator: Given i.i.d. samples from P_{S_\ell}, let \hat{P}_n be their empirical distribution. We estimate h(T_\ell) by \hat{h}_{SP}\triangleq h(\hat{P}_n \ast \gamma), which is computed only through the available resources: the samples and \gamma.
>
> (iv) MC Integration: Since \hat{P}_n is a discrete (known) distribution, \hat{P}_n \ast \gamma is a *known* n-mode Gaussian mixture with centers at the samples, and \hat{h}_{SP} equals the entropy of this mixture. This entropy (the aforementioned d-dimensional integral) has no closed-form expression, but since the Gaussian mixture is known (we know both \hat{P}_n and \gamma), we can efficiently compute its entropy by MC integration.
>
> We hope this clarifies our two-step process (first estimation and then computation) and that the estimator is \hat{h}_{SP}, and not the MC integrator. That this was unclear from the paper suggests the presentation might have been lacking; we will invest efforts in making the final version crystal clear.

---

> ### Author Response · Authors · 2018-11-09
> **Response to Reviewer 1 (part 3)**
>
> "Regarding the first experiment, could the authors clarify how per unit and per entire layer compression estimation differs?"
>
> The mechanics of the estimation are the same: in both cases we use H(Bin(T_\ell)) estimator. The difference is that in the entire layer case we are looking at the joint distribution P_{T_\ell}, while in the per unit case we are looking at the marginal distribution P_{T_\ell(k)}. In the case of the full layer, the output of each unit is discretized into two bins (as described in the caption of Fig. 7), while for the per-unit measurements we tested bins with widths in {10^-5, 10^-4, 10^-3, 10^-2, 0.1, 0.2, 0.3}, and found consistent results for bin sizes in [10^-4, 0.2].
>
> One problem with the per unit computation is that we then have d_\ell mutual information trajectories, one for each unit k\in[1:d_\ell], over the course of training that must be summarized. We summarize them by computing a linear regression that predicts I(X;T_\ell(k)) from the training epoch, t, for each unit k, and then looking at the distribution of the slopes of the regressors. Because most of the slopes are negative, this shows a trend that I(X;T_\ell(k)) decreases as t increases, which suggests that clustering is occurring.
>
> What we are most interested in is characterizing the clustering of samples in the representation computed by an entire layer. However, because differential entropy estimation has sample complexity exponential in dimension, we can only use I(X; T_\ell) to characterize clustering for small numbers of hidden units. The single-dimension results are suggestive that clustering is occurring, even though we cannot show it on the full layer.
>
>
> "Also, in my opinion, more clustered representations seem to indicate that the mutual information with the output increases. Could the authors comment on how the noise levels in this particular version of a stochastic network affects the mutual information with the output and the clustering? Do more clustered representations lead to increased mutual information of the layer with the output?"
>
> We kindly request a clarification here as to whether the reviewer meant the `output' of the network or the target (true) label?
>
> We ask this since we believe the mutual information between the hidden layer and the true label I(Y;T_\ell) is more informative, while the DNN's output does not necessarily equal the true label. While the current paper focuses on studying the behavior of I(X:T_\ell), we have a few comments regarding I(Y;T_\ell). First, we think that larger values of I(Y;T_\ell) are more related to having a good separation between the classes rather than to clustering itself. One way to see this is to note that for the last hidden T_{L-1}, I(Y;T_{L-1}) is essentially the cross-entropy loss. Studying I(Y;T_\ell) is on our research agenda and, in fact, we have just begun to explore it.
>
>
> "I found it fairly difficult to summarize the experimental contribution after the first read. I think the presentation and summary after each experiment could be improved and made more reader friendly. For example, the authors could include a short section before the experiments stating their hypothesis and pointing to the experiment/figure number supporting their hypothesis."
>
> Thanks for this helpful suggestion. We will revise our paper accordingly, while also trying to respect the ICLR page limit.

---

### Official Review · AnonReviewer2 · 2018-11-04
**Clarification of Compression Phrase in Information Bottleneck theory of DNNs**

**Rating:** 7
**Confidence:** 4

**Review:**

This paper provides a principled way to examine the compression phrase, i.e, I(X;T) in deep neural networks. To achieve this, the authors provides an theoretical sounding entropy estimator to estimate mutual information.  Empirically, the paper did observe this compression phrase across both synthetic and real-world data and relates this compression behavior with geometric clustering.

Pros:
- The paper is well-written and easy to understand.
- The framework for analyzing the mutual information in DNNs is theoretically sounding and robust.
- The finding of connecting clustering with compression is novel and inspiring.

Questions:
- The main concern of the paper is its conclusion. While the experiments in the paper did show the mutual information goes down as the clustering effect enhanced, it only means `clustering` and `compression` are correlated; but the paper claims `clustering` is the source of `compression`, i.e., `clustering` leads to `compression`. This conclusion is problematic. For example, looking at Figure 5(a), as the mutual information goes down from epoch 28 to epoch 8796, not only the clustering gets enhanced, but also the loss is going down. Thus, alternatively, one can also argue the loss (i.e., `relevance`) is the cause of `compression` instead of `clustering`. From another aspect, the effect of `clustering` is also related to the loss, i.e., it is the loss function that pushes the points of the same class to be closer; then, even if the direct cause of `compression` is `clustering`, the root cause might still be the loss (i.e., `relevance`).
- In Figure 5(a). Why the mutual information increases from epoch 80 - epoch 541? Also, it seems that the test loss increases as the I(X;T) decreases from epoch 541 to epoch 8796. This seems to be counter-intuitive to the claim that "lower I(X;T) implies higher generalization ability". Can you explain this phenomenon?

[UPDATE] the authors address my concerns in a detailed way, and the updated revision is rather robust, therefore, I decide to change my score to accept.

---

> ### Author Response · Authors · 2018-11-09
> **Response to Reviewer 2**
>
> "The main concern of the paper is its conclusion. While the experiments in the paper did show the mutual information goes down as the clustering effect enhanced, it only means 'clustering' and 'compression' are correlated; but the paper claims 'clustering' is the source of 'compression', i.e., 'clustering' leads to 'compression'. This conclusion is problematic. For example, looking at Figure 5(a), as the mutual information goes down from epoch 28 to epoch 8796, not only the clustering gets enhanced, but also the loss is going down. Thus, alternatively, one can also argue the loss (i.e., 'relevance') is the cause of 'compression' instead of 'clustering'. From another aspect, the effect of 'clustering' is also related to the loss, i.e., it is the loss function that pushes the points of the same class to be closer; then, even if the direct cause of 'compression' is 'clustering', the root cause might still be the loss (i.e., 'relevance')."
>
> We agree with R2 that, ultimately, all of the dynamics we observe are driven by the training algorithm working to reduce the training loss. However, our results show that clustering is the immediate cause of compression, i.e., whenever information compression occurs it is due to clustering. Furthermore, we have shown far more than simple correlation between compression and clustering for the following two reasons:
>
>
> 1. The analysis of information transmission over an additive white Gaussian noise (AWGN) channel in Section 4 shows directly how moving the representations of training samples closer together (that is, clustering them) causes a reduction in I(X;T_\ell) (that is, compression).
>
>
> 2. Our analysis of a minimal example in Section 4 illustrates the causal relationship between clustering and compression in low dimensions, where human geometric intuitions are reliable.
>
>
> We also stress that our results are incompatible with a claim that *reduction in loss* is correlated with compression. Multiple different trends are observable in relation to loss and compression: while there are instances where reduction in loss and compression simultaneously occur, there are other instances when loss decreases but mutual information rises. Two examples of the latter are the following:
>
>
> 1. Fig. 5(a), between epochs 80 and 541, shows that the training loss decreases while I(X;T_\ell) increases. The scatter plots show why: the representations of the training samples in layer 5 are rearranged from compact clusters into a more uniform (spread out) tube.
>
>
> 2. Similarly, a comparison of the results in Fig. 5(a) and 5(b) shows that the introduction of Parseval regularization does not interfere with the reduction of training loss, but it does eliminate compression and the mutual information keeps increasing from epoch roughly 500 and until the end of training. The reason why compression is eliminated is that Parseval regularization suppresses the network's ability to saturate all its units and form the tight clusters at the corners of the cube as it did in the unregularized experiment from Fig. 5(a). Indeed, the final constellation of internal representations in Fig. 5(b) (see scatter plot for epoch 7230) has no tight clusters. This stands in accordance with the claimed relations between clustering and compression.
>
>
> "In Figure 5(a). Why the mutual information increases from epoch 80 - epoch 541?"
>
> The constellation of training samples in layer 5 at epoch 541 is an elongated tube, while at epoch 80 it is a set of compact clusters. The increase in I(X;T_\ell) is consistent with our explanation that compression is caused by clustering: the tube in epoch 541 is more spread out than the clusters in epoch 80.
>
> "Also, it seems that the test loss increases as the I(X;T) decreases from epoch 541 to epoch 8796. This seems to be counter-intuitive to the claim that 'lower I(X;T) implies higher generalization ability'. Can you explain this phenomenon?"
>
>
> We emphasize that we never claimed that lower I(X;T_\ell) values imply better generalization: this claim was made in (Shwartz-Ziv & Tishby, 2017). In fact, as the reviewer points out, our empirical results indicate that this is not always the case. Fig. 5(a) is an excellent example of that, showing that too much clustering/compression probably results in overfitting, which is why the test loss grows towards the end. For practical purposes, early stopping would probably have been helpful here. However, since we are not concerned with attaining the best possible classification results, but rather understanding compression, we ran the training beyond the optimal stopping point. We hope this clarifies our stance regarding the relation between compression and generalization, and we will add a discussion in the revision to this effect.

---

### Author Response · Authors · 2018-11-09
**New revision**

We've uploaded a revision that changes Fig. 4(d) to show mutual information as a function of epochs, per Reviewer 3's request.  We are working on additional revisions recommended by the reviewers, and will upload another revision the week of Nov. 12.

---

### Author Response · Authors · 2018-11-15
**Uploaded revision 2**

We have uploaded a new revision of our paper that addresses the concerns raised by the reviewers. To make it clear what has changed, new text is highlighted in blue. The highlighting will, of course, be removed in the final revision of the paper.
To be specific, we have added material in response to the following concerns:
1. R2 asked about the apparent disconnect between compression and generalization performance (specifically loss on the test set) seen in Fig. 5(a) and (b).  We discuss this issue on page 7.
2. R1 asked whether Theorem 1 can be used to set the number of samples used in the sample propagation estimator of I(X;T) and R3 asked how the noise variance \beta^2 is chosen. We discuss these issues in Remark 1 on page 5.
3. R1 was concerned that the sample-propagation estimator is simply Monte Carlo integration. We have added text clarifying the distinction between the sample-propagation estimator, which casts the estimation of differential entropy in our noisy DNNs as estimation of the differential entropy of a known Gaussian mixture model, and Monte Carlo integration, which is used to numerically evaluate the differential entropy of the GMM, on pages 4 and 5.
4. R1 suggested that we "include a short section before the experiments stating their hypothesis and pointing to the experiment/figure number supporting their hypothesis." Instead, we have added a bit more text at the beginning of Section 5 (page 6) discussing the goals of our experiments, and added a summary of our findings like what R1 suggested at the end of Section 5 (page 9).
5. We replaces Fig. 4(d) with one that shows I(X;T) as a function of epoch (instead of weight) and different values of \beta, per R3's request.
6. Unless one of the reviewers or the AC objects, we currently plan to leave Supplement 10 intact, except that we will add a non-anonymized citation of our theory paper.

The additional material has increased the length of the main paper, excluding references, from 8 pages to 9, which is still within the ICLR page limits.

We thank the reviewers for their helpful feedback, and we hope that they will find that the revised paper addresses their concerns.

---

### Author Response · Authors · 2018-11-19
**Uploaded revision 3**

We are grateful to Reviewer 1 for the feedback on our second revision of the paper. Based on the concerns that R1 has raised about Remark 1, we have slightly expanded it to more closely follow the explanation we included in our response to R1.

We look forward to feedback from the other reviewers on our revision, and any additional recommendations that R1 might have.```

---

### Meta-Review · Area_Chair1 · 2018-12-16
**Contributes to resolving debate on compression in neural networks**

**Confidence:** 4
**Recommendation:** Reject

**Metareview:**

This paper studies the compression aspect of the information bottleneck. It seeks to clarify a debate about the evolution of mutual information between inputs and representations during training in neural networks. The paper discusses numerous ideas and techniques and arrives at valuable conclusions.

A concern is that parts of the paper (theoretical parts) are intended for a separate paper, and are included in the paper only for reference. This means that the actual contribution of the present paper is mostly on the experimental part. Nonetheless, the discussion derived from the theory and experiments seem valuable in the ongoing discussion of this topic. In any case, I encourage the authors to make efforts to obtain a transparent separation of the different pieces of work.

A concern was raised that the current paper mainly addresses a discussion that originated in a paper that has not passed peer review. On the other hand, this discussion does occupy many researchers and justifies the analysis, even if the originating paper has not been published in a peer reviewed format.

All reviewers are confident in their assessment. Two of them regard the paper positively and one of them regards the paper as ok, but not good enough, with main criticism in relation to the points discussed above.

Although the paper is in any case very good, unfortunately it does not reach the very high bar for acceptance at this ICLR.